# Bayesian Nonparametrics Meets Data-Driven Distributionally Robust Optimization

**Nicola Bariletto**
Department of Statistics and Data Sciences
The University of Texas at Austin
Austin, TX 78712
nicola.bariletto@utexas.edu

**Nhat Ho**
Department of Statistics and Data Sciences
The University of Texas at Austin
Austin, TX 78712
minhnhat@utexas.edu

## Abstract

Training machine learning and statistical models often involves optimizing a data-driven risk criterion. The risk is usually computed with respect to the empirical data distribution, but this may result in poor and unstable out-of-sample performance due to distributional uncertainty. In the spirit of distributionally robust optimization, we propose a novel robust criterion by combining insights from Bayesian nonparametric (i.e., Dirichlet process) theory and a recent decision-theoretic model of smooth ambiguity-averse preferences. First, we highlight novel connections with standard regularized empirical risk minimization techniques, among which Ridge and LASSO regressions. Then, we theoretically demonstrate the existence of favorable finite-sample and asymptotic statistical guarantees on the performance of the robust optimization procedure. For practical implementation, we propose and study tractable approximations of the criterion based on well-known Dirichlet process representations. We also show that the smoothness of the criterion naturally leads to standard gradient-based numerical optimization. Finally, we provide insights into the workings of our method by applying it to a variety of tasks based on simulated and real datasets.

## 1   Introduction

In machine learning and statistics applications, several quantities of interest solve the optimization problem

$$\min_{\theta \in \Theta} \mathcal{R}_{p_*}(\theta),$$

where $\mathcal{R}_p(\theta) := \mathbb{E}_{\xi \sim p}[h(\theta, \xi)]$ is the expected risk associated to decision $\theta$, under cost function $h(\theta, \xi)$ (measurable in the argument $\xi$) and given that the distribution of the $(\Xi, \mathscr{B}(\Xi))$-valued data $\xi$ is $p$.[1] For instance, if we are dealing with a supervised learning task where $\xi = (x, y) \in \mathbb{R}^{m-1} \times \mathbb{R}$, $h(\theta, \xi)$ is usually a loss function $\ell(f_\theta(x), y)$ quantifying the cost incurred in predicting $y$ with $f_\theta(x)$ – here the decision variable is $\theta$, which parametrizes the function $f_\theta : \mathbb{R}^{m-1} \to \mathbb{R}$. For the rest of the paper, we assume $\Xi \subseteq \mathbb{R}^m$, $\Theta \subseteq \mathbb{R}^d$ and $h : \Theta \times \Xi \to [0, K]$ for some $K < \infty$.[2]

In most cases of interest the true data-generating process $p_*$ is unknown, and only a sample $\boldsymbol{\xi}^n = (\xi_1, \ldots, \xi_n)$ from it is available. The most popular solution is to approximate $p_*$ by the empirical distribution $p_{\boldsymbol{\xi}^n}$, and optimize $\mathcal{R}_{p_{\boldsymbol{\xi}^n}}(\theta) \equiv n^{-1} \sum_{i=1}^n h(\theta, \xi_i)$. However, especially for small sample sizes $n$ and complex data-generating mechanisms, this can result in poor out-of-sample performance, leading to the need for robust alternatives. A flourishing literature on Distributionally Robust

---

[1]Given a topological space $(A, \mathcal{T})$, we denote by $\mathscr{B}(A)$ the Borel $\sigma$-algebra generated by $\mathcal{T}$.

[2]Note that, under mild regularity (e.g., continuity) conditions on $h$, its boundedness is ensured, for instance, by the frequently assumed compactness of $\Xi$ and $\Theta$.

38th Conference on Neural Information Processing Systems (NeurIPS 2024).

Optimization (DRO) has provided several methods in that direction [though not always with data-driven applications as the primary focus; see 35, for a recent exhaustive review of the field]. A prominent approach is the min-max DRO (mM-DRO) one, whereby a worst-case criterion over an ambiguity[3] set of plausible distributions is minimized [19, 2, 3, 10, 46, 11]. Recent notable results involve the study of mM-DRO problems where the ambiguity set is defined as a Wasserstein ball of probability measures centered at the empirical distribution [29, 24].

**Contribution.** Differently from the mM-DRO paradigm, we propose a distributionally robust procedure based on the minimization of the following criterion:

$$V_{\boldsymbol{\xi}^n}(\theta) := \int_{\mathscr{P}_{\Xi}} \phi(\mathcal{R}_p(\theta)) Q_{\boldsymbol{\xi}^n}(\mathrm{d}p), \tag{1}$$

where $\phi : [0, K] \to \mathbb{R}_+$ is a *continuous*, *convex* and *strictly increasing* function, and $Q_{\boldsymbol{\xi}^n}$ is a Dirichlet process posterior conditional on $\boldsymbol{\xi}^n$ [13].

As we show below, our proposal brings together insights from two well-established strands of literature – decision theory under ambiguity and Bayesian nonparametric statistics, – contributing in a novel way to the field of data-driven distributionally robust optimization. As we establish throughout the article, among the key advantages of the criterion are: (i) its favorable statistical properties in terms of probabilistic finite-sample and asymptotic performance guarantees; (ii) the availability of tractable approximations that are easy to optimize using standard gradient-based methods; and (iii) its ability to both improve and stabilize the out-of-sample performance of standard learning methods.

The rest of the paper is organized as follows. In Section 2, we motivate the formulation in Equation (1) by providing a concise overview of decision theory under ambiguity and its connections to Bayesian statistics and regularization. In Section 3, we study the statistical properties of procedures based on $V_{\boldsymbol{\xi}^n}$. In Section 4, we propose and study tractable approximations for $V_{\boldsymbol{\xi}^n}$ based on the theory of DP representations. In Section 5, we highlight the robustness properties of our method by applying it to a variety of learning tasks based on real and simulated data. Section 6 concludes the article. Proofs of theoretical results and further background are provided in Appendices A and B, respectively, while in Appendix C we discuss how the smoothness of the proposed criterion yields straightforward gradient-based optimization, and present more details on the numerical experiments. Code to replicate our experiments can be found at the folllowing link: `https://github.com/nbariletto/BNP_for_DRO`.

## 2    Decision Theory and Bayesian Statistics

Following a long-standing tradition in Bayesian statistics and decision theory [37], the distributional uncertainty on the data-generating process $p_*$ can be dealt with by defining a prior $Q$ over it. We point out that this Bayesian approach contrasts with the classical one, where a prior would typically be placed directly on the parameter $\theta$, whose data-driven optimal value would be determined as a function (e.g., the mean or mode) of the resulting posterior distribution. In this new framework, instead, the parameter $\theta$ is treated as a variable to be optimized, while the prior is assigned to the entire data-generating process. This perspective allows us to incorporate several valuable Bayesian concepts, as we will clarify throughout the paper, while preserving the flexibility of the original optimization-based learning framework. Notably, $\theta$ does not need to be interpreted as the parameter of a full generative model, as would be required in a classical Bayesian setting. Instead, it can represent a vector of parameters associated with a generic, possibly complex loss function, such as those employed in modern deep learning architectures.

Specifically, our Bayesian approach is equivalent to modeling the observed data $\boldsymbol{\xi}^n = (\xi_1, \ldots, \xi_n)$ as exchangeable with de Finetti measure $Q$:

$$\xi_i \mid p \overset{\text{iid}}{\sim} p, \quad i = 1, \ldots, n,$$
$$p \sim Q.$$

Due to the stochasticity of $p$, $\mathcal{R}_p(\theta)$ is itself a random variable, and a sensible procedure is to maximize its posterior expectation. Let $Q_{\boldsymbol{\xi}^n}$ be the posterior law of $p$ conditional on the sample $\boldsymbol{\xi}^n$.

---

[3]Throughout the paper, we adopt the terms "ambiguity" and "uncertainty" interchangeably.

Then, one solves the following problem:

$$\min_{\theta \in \Theta} \mathbb{E}_{p \sim Q_{\boldsymbol{\xi}^n}}[\mathcal{R}_p(\theta)] = \min_{\theta \in \Theta} \int_{\mathscr{P}_\Xi} \int_\Xi h(\theta, \xi) p(\mathrm{d}\xi) Q_{\boldsymbol{\xi}^n}(\mathrm{d}p) = \min_{\theta \in \Theta} \mathbb{E}_{\xi \sim p(\cdot|\boldsymbol{\xi}^n)}[h(\theta, \xi)],$$

where $\mathscr{P}_\Xi$ denotes the space of probability measures on $\mathscr{B}(\Xi)$ endowed with the Borel $\sigma$-algebra $\mathscr{B}(\mathscr{P}_\Xi)$ generated by the topology of weak convergence, while $p(\mathrm{d}\xi|\boldsymbol{\xi}^n) := \int_{\mathscr{P}_\Xi} p(\mathrm{d}\xi) Q_{\boldsymbol{\xi}^n}(\mathrm{d}p)$ denotes the posterior predictive distribution. In sum, within this general Bayesian framework, the data-driven problem reduces to minimizing $h(\theta, \xi)$ averaged w.r.t. the posterior predictive distribution, i.e., $\min_{\theta \in \Theta} \mathcal{R}_{p(\cdot|\boldsymbol{\xi}^n)}(\theta)$.

## 2.1 The Dirichlet Process

A natural choice is to model the prior $Q$ as a Dirichlet process (DP), and $Q_{\boldsymbol{\xi}^n}$ is then a DP posterior. First proposed by [13], the DP is the cornerstone nonparametric prior over spaces of probability measures. Its specification involves a *concentration parameter* $\alpha > 0$ and a *centering probability measure* $p_0$. Intuitively, the DP is characterized by the following finite-dimensional distributions: $p \sim \mathrm{DP}(\alpha, p_0)$ implies $(p(A_1), \ldots, p(A_k)) \sim \mathrm{Dirichlet}(\alpha p_0(A_1), \ldots, \alpha p_0(A_k))$ for any finite measurable partition $\{A_1, \ldots, A_k\}$ of $\Xi$.[4] A key property of the DP is its almost sure discreteness, which allows to write $p \stackrel{\mathrm{d}}{=} \sum_{j \geq 1} p_j \delta_{\xi_j}$ (where probability weights and atom locations are independent). Moreover, the DP is conjugate with respect to exchangeable sampling. In our case, this means

$$p \sim \mathrm{DP}(\alpha, p_0) \Rightarrow Q_{\boldsymbol{\xi}^n} = \mathrm{DP}\left(\alpha + n, \frac{\alpha}{\alpha + n} p_0 + \frac{n}{\alpha + n} p_{\boldsymbol{\xi}^n}\right).$$

That is, conditional on the sample $\boldsymbol{\xi}^n$, $p$ is again a DP with larger concentration parameter $\alpha + n$ and centered at the predictive distribution $p(\cdot|\boldsymbol{\xi}^n) := \frac{\alpha}{\alpha+n} p_0 + \frac{n}{\alpha+n} p_{\boldsymbol{\xi}^n}$. The latter is a compromise between the prior guess $p_0$ and the empirical distribution $p_{\boldsymbol{\xi}^n}$, and the balance between the two is determined by the relative size of $\alpha$ and $n$. The predictive distribution is also related to the celebrated Blackwell-MacQueen Pólya urn scheme (or Chinese restaurant process) to draw an exchangeable sequence $(\xi_i)_{i \geq 1}$ distributed according to $p \sim \mathrm{DP}(\alpha, p_0)$: Draw $\xi_1 \sim p_0$ and, for all $i > 1$ and $\ell < i$, set $\xi_i = \xi_\ell$ with probability $(\alpha + j - 1)^{-1}$, else (i.e., with probability $\alpha(\alpha + j - 1)^{-1}$) draw $\xi_i \sim p_0$ [4].

Given the large support of $Q_{\boldsymbol{\xi}^n}$, which consists of all probability measures whose support is included in that of $p(\cdot|\boldsymbol{\xi}^n)$ [28], the DP is a reasonable and tractable option to mitigate misspecification concerns. Then, leveraging the mentioned expression for the DP predictive distribution, the problem specializes to

$$\min_{\theta \in \Theta} \left\{ \frac{n}{\alpha + n} \mathcal{R}_{p_{\boldsymbol{\xi}^n}}(\theta) + \frac{\alpha}{\alpha + n} \mathcal{R}_{p_0}(\theta) \right\} \tag{2}$$

[see also 27, 45]. In practice, adopting the above Bayesian approach amounts to introducing a regularization term depending on the prior centering distribution $p_0$. Compared to the simple empirical risk $\mathcal{R}_{p_{\boldsymbol{\xi}^n}}(\theta)$, this type of criterion displays lower variance (because $\mathcal{R}_{p_0}(\theta)$ is non-random) at the cost of some additional, asymptotically-vanishing bias w.r.t. the theoretical criterion $\mathcal{R}_{p_*}(\theta)$. We also note that such bias can be attenuated in finite samples as long as the prior guess $p_0$ and the true data-generating process $p_*$ are close enough in terms of the difference $|\mathcal{R}_{p_0}(\theta) - \mathcal{R}_{p_*}(\theta)|$.[5]

**Connections to Regularization in Linear Regression.** One of the most ubiquitous data-driven learning tasks is linear regression [38, 7]. It is well-known that, in this setting, coefficient estimation (e.g., via maximum likelihood or least squares) can be framed as a minimization problem of the sample average of the squared loss function. It turns out that, applying the Bayesian regularized approach (2), an interesting equivalence with standard regularization techniques such as Ridge [21] and LASSO [41] emerges.

**Proposition 2.1.** *Let* $h(\theta, (y, x)) = (y - \theta^\top x)^2$. *Then, denoting* $\lambda_{\alpha,n} := \alpha/n$, *the following equivalences hold:*

---

[4]Also, $\mathbb{E}[p(A)] = p_0(A)$ and $\mathbb{V}[p(A)] = (1 + \alpha)^{-1} p_0(A)(1 - p_0(A))$ for any $A \in \mathscr{B}(\Xi)$, justifying the names of $\alpha$ and $p_0$.

[5]In practice, the prior guess $p_0$ can be leveraged to incorporate features of the underlying process that the researcher suspects to hold (e.g., in regression applications, sparsity). See also Section 5.

*1. If $p_0 = \mathcal{N}(0, I)$, then $\hat{\theta}$ solving (2) implies that it solves*

$$\min_{\theta \in \Theta} \left\{ \frac{1}{n} \sum_{i=1}^{n} h(\theta, \xi_i) + \lambda_{\alpha,n} \|\theta\|_2^2 \right\};$$

*2. If $V = \mathrm{diag}(|\theta_1|^{-1}, \ldots, |\theta_{d-1}|^{-1})$ and $p_0 = \mathcal{N}(0, V)$, then $\hat{\theta}$ solving (2) implies that it solves*

$$\min_{\theta \in \Theta} \left\{ \frac{1}{n} \sum_{i=1}^{n} h(\theta, \xi_i) + \lambda_{\alpha,n} \|\theta\|_1 \right\}.$$

Proposition 2.1 is insightful because it highlights a novel Bayesian interpretation of Ridge and LASSO linear regression. In fact, it is well known that both methods are equivalent to maximum-a-posteriori estimation of regression coefficients when the latter are assigned either a normal or a Laplace prior. In our setting, instead of a parametric prior on the regression coefficients, we place a nonparametric one on the joint distribution of the response and the covariates. The degree of regularization, then, is naturally guided by the prior confidence parameter $\alpha$ and the sample size $n$. We also note that sparsity is only one of the possible data-generating features one might want to enforce in regularized estimation,[6] and the nonparametric Bayesian approach offers greater flexibility, compared to Ridge and LASSO, to incorporate such patterns by specifying the prior expectation $p_0$ of the joint response-covariate distribution.

## 2.2 Ambiguity Aversion

As we just showed, adopting a traditional Bayesian framework, uncertainty about the model $p$ is resolved by using the posterior $Q_{\boldsymbol{\xi}^n}$ to directly average $p$ out. This procedure, however, does not take into account the (partly) subjective nature of the beliefs encoded in $Q_{\boldsymbol{\xi}^n}$, and the aversion to this that a statistical decision maker (DM) might have. In fact, the result of the procedure is that the DM ends up minimizing the expected risk, where the average is taken according to the predictive distribution. In practice, then, the latter is put on the same footing as an objectively known probability distribution, such as the true model.

This issue has been thoroughly studied and addressed in the economic decision theory literature [18, 6]. In that context, the economic DM faces an analogous expected utility maximization problem $\max_{\theta \in \Theta} \mathbb{E}_{\xi \sim p}[u(\theta, \xi)]$ (e.g., to allocate her capital to a portfolio of investments subject to random economic shocks $\xi$). However, she does not possess enough objective information to pick one single model of the world $p$, but deems a larger set of models plausible. One possibility, then, is that the DM forms a second-order belief (e.g., a prior $Q$) over such set, and resolves uncertainty by directly averaging expected utility profiles $\mathbb{E}_{\xi \sim p}[u(\theta, \xi)]$ w.r.t. $p \sim Q$.

Just like in our data-driven problem, however, direct averaging does not account for ambiguity aversion. [23] proposed and axiomatized a tractable "Smooth Ambiguity Aversion" (SmAA) model, whereby second-order averaging is preceded by a deterministic transformation $\phi$ inducing uncertainty aversion via its curvature: The DM optimizes $\int_{\mathscr{P}_\Xi} \phi(\mathbb{E}_{\xi \sim p}[u(\theta, \xi)]) Q(\mathrm{d}p)$, and criterion (1) simply specializes the SmAA model to the data-driven case.[7] When optimization takes the form of minimization, ambiguity aversion is driven by the degree of convexity of $\phi$.[8] In particular, convexity encodes the DM's tendency to pick decisions that yield less variable expected loss levels across ambiguous probability models. To see this intuitively, examine the simple case when only two models, $p_1$ and $p_2$, are supported by $Q = \frac{1}{2}\delta_{p_1} + \frac{1}{2}\delta_{p_2}$. Consider two decisions $\theta_1$ and $\theta_2$ that, under $p_1$ and $p_2$, yield the expected risks marked on the horizontal axis of Figure 1. While $\int \mathcal{R}_p(\theta_1) Q(\mathrm{d}p) = \int \mathcal{R}_p(\theta_2) Q(\mathrm{d}p) = \mathcal{R}^*$, the convexity of $\phi$ implies

---

[6]For instance, one might have prior information on specific correlation patterns among covariates, which could be useful to incorporate in regression training with few data points.

[7]Our approach is also related to recent literature [48, 47, 40] exploring Bayesian ideas in the context of DRO. However, the cited works (i) focus on parametric priors and (ii) rely on either ambiguity sets or statistical risk measures to induce robustness. This is in contrast with our work, which (i) resorts to a more assumption-free nonparametric prior and (ii) leverages the highlighted robustness properties of the simple yet powerful convex transformation $\phi$.

[8]In the economic decision theory literature, as the DM usually maximizes a criterion (utility), convexity is replaced by concavity.

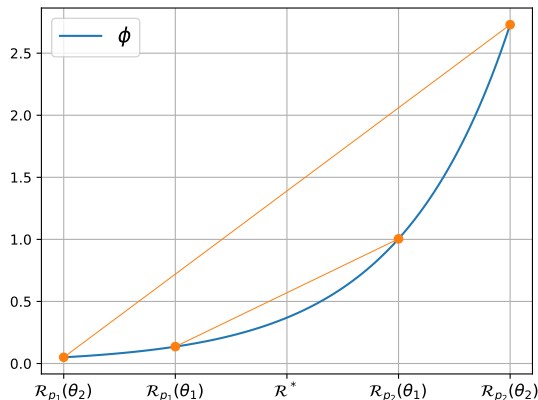

Figure 1: Graphical display of smooth ambiguity aversion at work. Although $\theta_1$ and $\theta_2$ yield the same loss $\mathcal{R}^*$ in $Q$-expectation, the ambiguity averse criterion favors the less variable decision $\theta_1$. Graphically, this is because the orange line connecting $\phi(\mathcal{R}_{p_1}(\theta_1))$ to $\phi(\mathcal{R}_{p_2}(\theta_1))$ lies (point-wise) below the line connecting $\phi(\mathcal{R}_{p_1}(\theta_2))$ to $\phi(\mathcal{R}_{p_2}(\theta_2))$.

$\int \phi(\mathcal{R}_p(\theta_1))Q(\mathrm{d}p) < \int \phi(\mathcal{R}_p(\theta_2))Q(\mathrm{d}p)$. That is, although $\theta_1$ and $\theta_2$ yield the same loss in $Q$-expectation, the ambiguity-averse criterion favors $\theta_1$ because it ensures less variability across uncertain distributions $p_1$ and $p_2$.

Interestingly, [5] showed that the SmAA model belongs to a general class of ambiguity-averse preferences, which admit a common utility function representation. For SmAA preferences with $\phi(t) = \beta \exp(\beta^{-1}t) - \beta$ (with $\beta > 0$ and under additional technical assumptions), this representation implies the equivalence of problem (1) with

$$\min_{\theta \in \Theta} \max_{P:P \ll Q_{\boldsymbol{\xi}^n}} \left\{ \mathbb{E}_{p \sim P}[\mathcal{R}_p(\theta)] - \beta\mathrm{KL}(P\|Q_{\boldsymbol{\xi}^n}) \right\},$$

where $\mathrm{KL}(\cdot\|\cdot)$ is the Kullback-Leibler divergence and $\ll$ denotes absolute continuity. The above result further clarifies the mechanism through which distributional robustness is induced: Intuitively, instead of directly averaging over $p \sim Q_{\boldsymbol{\xi}^n}$, one computes a worst-case scenario w.r.t. the mixing measure, penalizing distributions that are further away from the posterior – the latter acts as a reference probability measure. Moreover, in the limiting case $\beta \to 0$, the mM-DRO setup is recovered, with ambiguity set $\mathcal{C} = \{p \in \mathscr{P}_\Xi : \exists P \ll Q_{\boldsymbol{\xi}^n}, p = \int_{\mathscr{P}_\Xi} qP(\mathrm{d}q)\}$. In the other limiting case $\beta \to \infty$ (with the convention $0 \cdot \infty = 0$), the ambiguity neutral Bayesian criterion (2) is instead recovered.

## 3   Statistical Properties

In this section, we analyze the statistical properties of the criterion $V_{\boldsymbol{\xi}^n}(\theta)$, as a function of the sample size $n$. A first issue of interest, addressed in Proposition 3.1, is to study its asymptotic point-wise behavior (cf. [17], Corollary 4.17).

**Proposition 3.1.** *Let $\boldsymbol{\xi}^n$ be iid according to $p_*$ and $\xi \mapsto h(\theta, \xi)$ continuous for all $\theta \in \Theta$. Then, for all $\theta \in \Theta$,*

$$\lim_{n \to \infty} V_{\boldsymbol{\xi}^n}(\theta) = \phi(\mathcal{R}_{p_*}(\theta))$$

*almost surely.*

This ensures that, as more data are collected, the proposed criterion approaches, with probability 1, the true theoretical risk (up to the strictly increasing transformation $\phi$).

While point-wise convergence to the target ground truth is a first desirable property for any sensible criterion, it is not enough to characterize the behavior of the optimization's out-of-sample performance, nor the closeness of the optimal criterion value and the criterion optimizer(s) to their theoretical counterparts. In the following subsections, we study these properties both in the finite-sample regime and in the asymptotic limit $n \to \infty$.

**Finite-Sample Guarantees.**   Denote

$$\theta_n \in \arg\min_{\theta\in\Theta} V_{\boldsymbol{\xi}^n}(\theta), \quad \theta_* \in \arg\min_{\theta\in\Theta} \mathcal{R}_{p_*}(\theta),$$

and we assume the above sets of minimizers to be non-empty throughout the article. In finite-sample analysis, a first question of interest is whether probabilistic performance guarantees hold for the robust criterion optimizer $\theta_n$. In our setting, one can naturally measure performance by the narrowness of the gap between $\mathcal{R}_{p_*}(\theta_n)$ and $\mathcal{R}_{p_*}(\theta_*)$. As we clarify later, Lemma 3.2 is a first step towards establishing this type of guarantees.

**Lemma 3.2.** *Let $\phi$ be twice continuously differentiable on $(0, K)$, with $M_\phi := \sup_{t\in(0,K)} \phi'(t) < +\infty$ and $\gamma_\phi^* := \sup_{t\in(0,K)} \gamma_\phi(t) < +\infty$, where $\gamma_\phi(t) := \phi''(t)/\phi'(t) \geq 0$. Then*

$$\sup_{\theta\in\Theta} |V_{\boldsymbol{\xi}^n}(\theta) - \phi(\mathcal{R}_{p_*}(\theta))| \leq M_\phi \left[ \frac{n}{\alpha + n} \sup_{\theta\in\Theta} |\mathcal{R}_{p_{\boldsymbol{\xi}^n}}(\theta) - \mathcal{R}_{p_*}(\theta)| + \frac{\alpha}{\alpha + n} K + \frac{K^2}{2}\gamma_\phi^* \right].$$

Lemma 3.2 links the $\sup$ distance of the criterion $V_{\boldsymbol{\xi}^n}$ from the theoretical risk to three key objects:

1. The classical $\sup$ distance between the empirical and theoretical risk, $\sup_{\theta\in\Theta} |\mathcal{R}_{p_{\boldsymbol{\xi}^n}}(\theta) - \mathcal{R}_{p_*}(\theta)|$;

2. The $\sup$ distance between the theoretical risk and the risk computed w.r.t. the base probability measure $p_0$, $\sup_{\theta\in\Theta} |\mathcal{R}_{p_0}(\theta) - \mathcal{R}_{p_*}(\theta)|$. In fact, while in the formulation of Lemma 3.2 we bound such distance by $K$ (see the second addendum) to eliminate the dependence on the unknown but fixed $p_*$, one could equivalently replace $K$ by $\sup_{\theta\in\Theta} |\mathcal{R}_{p_0}(\theta) - \mathcal{R}_{p_*}(\theta)|$. This clarifies that, if $p_0$ is a good guess for $p_*$, i.e., if the above $\sup$ distance is small, adopting a Bayesian prior centered at $p_0$ can improve finite sample bounds;

3. The *Arrow-Pratt coefficient* $\gamma_\phi(t)$ of absolute ambiguity aversion. In the economic theory literature on decision-making under risk, this is a well-known concept measuring the degree of risk aversion of decision makers, with point-wise larger values of $\gamma_\phi$ corresponding to more risk aversion. See [23] for a discussion on the straightforward adaptation of this measure to the ambiguity (rather than risk) aversion setup we work in.

Most importantly, Lemma 3.2 allows us to prove the following Theorem, which yields the performance guarantees we are after.

**Theorem 3.3.** *For all $\delta > 0$*

$$\mathbb{P}[\phi(\mathcal{R}_{p_*}(\theta_n)) - \phi(\mathcal{R}_{p_*}(\theta_*)) \leq \delta]$$
$$\geq \mathbb{P}\left[ \sup_{\theta\in\Theta} |\mathcal{R}_{p_{\boldsymbol{\xi}^n}}(\theta) - \mathcal{R}_{p_*}(\theta)| \leq \frac{\alpha + n}{n}\left( \frac{\delta}{2M_\phi} - \frac{\alpha}{\alpha + n}K - \frac{K^2}{2}\gamma_\phi^* \right) \right].$$

Theorem 3.3 allows to obtain finite-sample probabilistic guarantees on the excess risk $\phi(\mathcal{R}_{p_*}(\theta_n)) - \phi(\mathcal{R}_{p_*}(\theta_*))$ via bounds on $\sup_{\theta\in\Theta} |\mathcal{R}_{p_{\boldsymbol{\xi}^n}}(\theta) - \mathcal{R}_{p_*}(\theta)|$. The latter is a well-studied quantity, and the sought bounds follow from standard results relying on conditions on the complexity of the function class $\mathcal{H} := \{h(\theta, \cdot) : \theta \in \Theta\}$, as measured by its Vapnik–Chervonenkis dimension, metric entropy, etc. We refer the reader to [44, 43] for a systematic treatment of the topic and specific useful results.

**Asymptotic Guarantees.**   So far, we have studied the finite-sample behavior of the out-of-sample performance of $\theta_n$. Another closely related type of results deals with the asymptotic limit of such performance, as well as with the convergence of optimum criterion values and optimizing parameters to their ground-truth counterparts. In this Subsection, attention is turned to theoretical results of this kind.

Finite-sample guarantees on $\sup_{\theta\in\Theta} |\mathcal{R}_{p_{\boldsymbol{\xi}^n}}(\theta) - \mathcal{R}_{p_*}(\theta)|$ are usually of the form

$$\mathbb{P}\left[ \sup_{\theta\in\Theta} |\mathcal{R}_{p_{\boldsymbol{\xi}^n}}(\theta) - \mathcal{R}_{p_*}(\theta)| \leq \delta \right] \geq 1 - \eta_n,$$

with $\sum_{n=1}^{\infty} \eta_n < \infty$. This implies (via a straightforward application of the first Borel-Cantelli Lemma) the almost sure vanishing of $\sup_{\theta \in \Theta} \left| \mathcal{R}_{p_{\boldsymbol{\xi}^n}}(\theta) - \mathcal{R}_{p_*}(\theta) \right|$. Thus, we include this as an assumption of the next Theorem. Moreover, we introduce a functional dependence of $\phi$ on $n$, and denote $\phi \equiv \phi_n$ accordingly.

**Theorem 3.4.** *Retain the assumptions of Lemma 3.2 and* $\lim_{n \to \infty} \sup_{\theta \in \Theta} \left| \mathcal{R}_{p_{\boldsymbol{\xi}^n}}(\theta) - \mathcal{R}_{p_*}(\theta) \right| = 0$ *almost surely. Moreover, assume that $\phi_n$ satisfies (1) $\lim_{n \to \infty} \gamma^*_{\phi_n} = 0$, (2) $\sup_{n \geq 1} M_{\phi_n} < \infty$, and (3) $\lim_{n \to \infty} \sup_{t \in [0,K]} |\phi_n(t) - t| = 0$. Then the next two almost sure limits hold:*

$$\lim_{n \to \infty} \mathcal{R}_{p_*}(\theta_n) = \mathcal{R}_{p_*}(\theta_*), \quad \lim_{n \to \infty} V_{\boldsymbol{\xi}^n}(\theta_n) = \mathcal{R}_{p_*}(\theta_*).$$

Theorem 3.4 is crucial because it ensures that, asymptotically, the excess risk vanishes and the finite-sample optimal value converges to the optimal value under the data generating process.[9]

*Remark* 3.5. From a design point of view, the type of $n$-dependent parametrization of $\phi$ required in Theorem 3.4 is sensible, as it is equivalent to adopting vanishing levels of ambiguity aversion (uniformly vanishing Arrow-Pratt coefficient) as the sample size grows – that is, as one obtains a more and more precise estimate of the true distribution $p_*$. Moreover, this assumption is in the spirit of the condition imposed on the radius of the Wasserstein ambiguity ball in [29], which is required to vanish as the sample size grows. Finally, it is easy to show that $\phi_n(t) = \beta_n \exp(\beta_n^{-1} t) - \beta_n$, for positive $\beta_n \to \infty$, satisfies the conditions of Theorem 3.4, and from now on we silently assume this form for $\phi_n$.

Finally, we leverage the above results to ensure the convergence of the sequence of optimizers $(\theta_n)_{n \geq 1}$ to a theoretical optimizer.

**Theorem 3.6.** *Let $\theta \mapsto h(\theta, \xi)$ be continuous for all $\xi \in \Xi$ and $\lim_{n \to \infty} \mathcal{R}_{p_*}(\theta_n) = \mathcal{R}_{p_*}(\theta_*)$ almost surely (e.g., as ensured in Theorem 3.4). Then, almost surely, $\lim_{n \to \infty} \theta_n = \bar{\theta}$ implies $\mathcal{R}_{p_*}(\bar{\theta}) = \mathcal{R}_{p_*}(\theta_*)$.*

## 4   Monte Carlo Approximation

In what follows, we fix a sample $\boldsymbol{\xi}^n$ and propose simulation strategies to estimate $V_{\boldsymbol{\xi}^n}(\theta)$. The latter, in fact, is analytically intractable due to the infinite dimensionality of $Q_{\boldsymbol{\xi}^n}$. To that end, we exploit a key representation of DPs first established by [39]: If $p \sim \mathrm{DP}(\eta, q)$, then $p \stackrel{\mathrm{d}}{=} \sum_{j=1}^{\infty} p_j \xi_j$, where $\xi_j \stackrel{\mathrm{iid}}{\sim} q$ and the sequence of weights $(p_j)_{j \geq 1}$ is constructed via a stick-breaking procedure based on iid $\mathrm{Beta}(1, \eta)$ samples (see Appendix B). Thus, for large enough integers $T$ and $N$, we propose the following Stick-Breaking Monte Carlo (SBMC) approximation for $V_{\boldsymbol{\xi}^n}(\theta)$:

$$\hat{V}_{\boldsymbol{\xi}^n}(\theta, T, N) := \frac{1}{N} \sum_{i=1}^{N} \phi \left( \sum_{j=0}^{T} p_{ij} h(\theta, \xi_{ij}) \right), \tag{3}$$

where, $T$ denotes the number of stick-breaking steps performed before truncating each Monte Carlo sample from the DP posterior $Q_{\boldsymbol{\xi}^n}$, while $N$ denotes the number of such samples. Algorithm 1 in Appendix B details the procedure, which essentially approximates the posterior DP via truncation and takes expectations accordingly.

*Remark* 4.1. We propose to truncate the stick-breaking procedure at some fixed step $T$. Another strategy would involve truncating it at a random step $T_i(\varepsilon) := \min \left\{ t \in \mathbb{N} : \prod_{j=1}^{t} p_{ij} \leq \varepsilon \right\}$ for some small $\varepsilon > 0$. This allows to directly control the approximation error at each Monte Carlo sample [30, 1], though it leads to simulated measures with supports of different cardinalities. For the sake of theory, we opt for the fixed-step/random-error approximation, though the random-step/fixed-error one is equally viable in practice.

*Remark* 4.2. On top of being a theory-based approximation for $V_{\boldsymbol{\xi}^n}(\theta)$, the criterion (3) can be interpreted as implementing a form of *robust Bayesian bootstrap*. Instead of directly averaging the risk $h(\theta, \cdot)$ with respect to the empirical distribution, we first obtain $N$ bootstrap samples of

---

[9]Using an analogous line of reasoning as in the proof of Theorem 3.4 (see Appendix A), we note that Lemma 3.2 and Theorem 3.3 can be easily adapted in to obtain finite sample bounds on $\sup_{\theta \in \Theta} |V_{\boldsymbol{\xi}^n}(\theta) - \mathcal{R}_{p_*}(\theta)|$ and $\mathcal{R}_{p_*}(\theta_n) - \mathcal{R}_{p_*}(\theta_*)$ depending on $\sup_{t \in [0,K]} |\phi(t) - t|$.

size $T$ from the predictive (which is a compromise between the empirical and the prior centering distributions), we weight observations according to the stick-breaking procedure, and finally take a grand average of the $\phi$-transformed weighted sums. This connection with the Bayesian bootstrap suggests the following alternative Multinomial-Dirichlet Monte Carlo (MDMC) version of $V_{\boldsymbol{\xi}^n}(\theta)$:

$$\frac{1}{N}\sum_{i=1}^{N}\phi\left(\sum_{j=1}^{T}w_{ij}h(\theta,\xi_{ij})\right),$$

where $(w_{i1},\ldots,w_{iT}) \overset{\text{iid}}{\sim} \text{Dirichlet}\left(T;\frac{\alpha+n}{T},\ldots,\frac{\alpha+n}{T}\right)$ and the atoms are iid according to the predictive (see Algorithm 2 in Appendix B).[10] For practical computation, we recommend using the MDMC approximation, as it tends to yield more balanced weights, compared to SBMC, even for low values of $T$.

With the following results, we ensure finite-sample and asymptotic guarantees on the closeness of optimization procedures based on the SBMC approximation versus the target $V_{\boldsymbol{\xi}^n}$.

**Lemma 4.3.** *Assume $\Theta$ is a bounded subset of $\mathbb{R}^d$ and, for all $\xi \in \Xi$, $\theta \mapsto h(\theta,\xi)$ is $c(\xi)$-Lipschitz continuous. Then, for all $T, N \geq 1$ and $\varepsilon > 0$,*

$$\sup_{\theta \in \Theta}\left|\hat{V}_{\boldsymbol{\xi}^n}(\theta,T,N) - V_{\boldsymbol{\xi}^n}(\theta)\right| \leq M_{\phi}K \times \left[\frac{\alpha+n}{\alpha+n+1}\right]^{T} + \varepsilon$$

*with probability at least*

$$1 - 2\left(\frac{32M_{\phi}C_T\text{diam}(\Theta)\sqrt{d}}{\varepsilon}\right)^d \times \left[\exp\left\{-\frac{3N\varepsilon^2}{4\phi(K)(6\phi(K)+\varepsilon)}\right\} + \exp\left\{-\frac{3N\varepsilon}{40\phi(K)}\right\}\right]$$

*for some constant $C_T > 0$.*

Heuristically, the bound in Lemma 4.3 is obtained by decomposing the left-hand side of the inequality into a first term depending on the truncation error induced by the threshold $T$, and a second term reflecting the Monte Carlo error related to $N$. Moreover, analogously to Lemma 3.2, Lemma 4.3 easily implies finite-sample bounds on the excess "robust risk" $V_{\boldsymbol{\xi}^n}(\hat{\theta}_n(T,N)) - V_{\boldsymbol{\xi}^n}(\theta_n)$, where $\hat{\theta}_n(T,N) \in \arg\min_{\theta \in \Theta} \hat{V}_{\boldsymbol{\xi}^n}(\theta,T,N)$. Another consequence is the following asymptotic convergence Theorem, whose proof is analogous to that of Theorem 3.4.

**Theorem 4.4.** *Under the same assumptions of Lemma 4.3, and if $\sup_{T\geq 1} C_T < \infty$ (see Appendix A for details on $C_T$),*

$$\lim_{T,N\to\infty}\sup_{\theta \in \Theta}\left|\hat{V}_{\boldsymbol{\xi}^n}(\theta,T,N) - V_{\boldsymbol{\xi}^n}(\theta)\right| = 0$$

*almost surely. Also, almost surely*

$$\lim_{T,N\to\infty}\hat{V}_{\boldsymbol{\xi}^n}(\hat{\theta}_n(T,N),T,N) = V_{\boldsymbol{\xi}^n}(\theta_n), \qquad \lim_{T,N\to\infty}V_{\boldsymbol{\xi}^n}(\hat{\theta}_n(T,N)) = V_{\boldsymbol{\xi}^n}(\theta_n).$$

In words, Theorem 4.4 ensures that, as the truncation and MC approximation errors vanish, the optimal approximate criterion value converges to the optimal exact one, and that the exact criterion value at any approximate optimizer converges to the exact optimal value. Also note that Theorems 3.4 and 4.4, when combined, provide guarantees on the convergence of $\hat{V}_{\boldsymbol{\xi}^n}(\hat{\theta}_n(T,N),T,N)$ (the empirical criterion one has optimized in practice) to $\mathcal{R}_{p_*}(\theta_*)$ (the theoretical optimal target) as the sample size increases and the DP approximation improves.

Finally, as a byproduct of Theorem 4.4, convergence of any approximate robust optimizer to an exact one is established as follows.[11]

**Theorem 4.5.** *Let $\theta \mapsto h(\theta,\xi)$ be continuous for all $\xi \in \Xi$. Moreover, assume*

$$\lim_{T,N\to\infty}V_{\boldsymbol{\xi}^n}(\hat{\theta}_n(T,N)) = V_{\boldsymbol{\xi}^n}(\theta_n)$$

*almost surely (e.g., as ensured above). Then, almost surely, $\lim_{T,N\to\infty}\hat{\theta}_n(T,N) = \bar{\theta}_n$ implies $V_{\boldsymbol{\xi}^n}(\bar{\theta}_n) = V_{\boldsymbol{\xi}^n}(\theta_n)$.*

---

[10]In the $\alpha = 0$ limit and setting $T = n$, the well-known "Bayesian bootstrap distribution" is recovered [17, see also Appendix B for further details].

[11]In Appendix A, we also present an asymptotic normality result for the approximate optimizer $\hat{\theta}_n(T,N)$ (Proposition A.3).

# 5    Experiments

We applied our robust optimization procedure to a host of simulated and real datasets, and we report results in this Section. Before proceeding, we notice that, given the finite approximations proposed in Section 4 and under mild regularity assumptions on the loss function $h$, the proposed robust criterion is amenable to standard gradient-based optimization procedures (see Appendix C for further details and an insightful interpretation of the gradient of our criterion as yielding *robustly weighted stochastic gradient descent steps*).

**Simulation Studies.**   We tested our method on three different learning tasks featuring a high degree of distributional uncertainty in the data generating process, and compared performance to the corresponding ambiguity neutral (i.e., simply regularized) and unregularized procedures. First, we performed a high-dimensional sparse linear regression simulation experiment. We simulated 200 independent samples of size $n = 100$ from a linear model with $d = 90$ features (moderately correlated with each other), only the first $s = 5$ of which have unitary positive marginal effect on the scalar response $y$. Second, we performed a simulation experiment on univariate Gaussian mean estimation in the presence of outliers. We simulated 200 independent samples of 13 observations, 10 of which come from a 0-mean Gaussian distribution and 3 from an outlier distribution, and tested the ability of the three methods to recover the true mean (i.e., 0). Third, we performed a simulation experiment on high-dimensional sparse logistic regression for binary classification. We set up a data-generating mechanism similar to the linear regression experiment, where a small subset of features linearly influence the log odds-ratio.

Appendix C collects further details on the above experiments as well as plots summarizing the results (see Figures 2, 3, and 4). All three experiments reveal the ability of our robust method to improve out-of-sample performance and estimation accuracy in two ways, i.e., (i) by yielding good results on average, and especially (ii) by reducing performance variability. The latter is a key robustness property that our method is designed to achieve.

**Real Data Applications.**   We tested our method on three diverse real-world datasets. In the first study, we applied our method to predict diabetes development based on a host of features, as collected in the popular and public Pima Indian Diabetes dataset. Because the outcome is binary, we used logistic regression as implemented (i) with our robust method, (ii) with $L_1$ regularization, and (iii) in its plain, unregularized version. We selected hyperparameters via cross-validation and tested the out-of-sample performance of the three methods applied to disjoint batches of training observations to assess the methods' performance variability. As we report in Appendix C, our robust method outperforms both alternatives on average and does significantly better in reducing variability.

We performed two further studies on linear regression applied to two popular UCI Machine Learning Repository datasets: The Wine Quality dataset [8] and the Liver Disorders dataset [15]. Similarly to the first study, we compared the performance of our method to OLS (unregularized) estimation and $L_1$-penalized (LASSO) regression. After cross-validation for parameter selection, we train the models multiple times on separate batches of data and compute out-of-sample performance on a large held-out set of observations. As the results reported in Appendix C show, also in these settings our robust DP-based method performs better than the alternatives both on average and especially in terms of lower variability. Taken together, the experimental results described in this Section corroborate empirically the robustness properties of the proposed criterion, as examined theoretically throughout the paper.

# 6    Discussion

The paper tackled the problem of optimizing a data-driven criterion in the presence of distributional uncertainty about the data-generating mechanism. To mitigate the underperformance of classical methods, we introduced a novel distributionally robust criterion, drawing insights from Bayesian nonparametrics and a decision-theoretic model of smooth ambiguity aversion. We established connections with standard regularization techniques, including Ridge and LASSO regression, and theoretical analysis revealed favorable finite-sample and asymptotic guarantees on the performance of the robust procedure. For practical implementation, we presented and examined tractable approximations of the criterion, which are amenable to gradient-based optimization. Finally, we applied our method

to a variety of simulated and real datasets, offering insights into its practical robustness properties. Naturally, our work presents some limitations that give rise to interesting directions for future research. In particular, we note the need for a deeper examination of the model workings in terms of (i) its parameter configuration and (ii) its broader application to general learning tasks (e.g., when the loss function is adapted to accommodate deep learning architectures). Moreover, our method, as many others in the distributional robustness literature, is only suited to process homogeneously generated (e.g., iid or exchangeable) data, leaving room to explore extensions to more complex dependence structures. Finally, we highlight that our study offers prospects for investigating connections among such varied yet interconnected strands of literature as optimization, decision theory, and Bayesian statistics.

## Acknowledgements

Both authors would like to thank Khai Nguyen for useful advice on coding implementation. NB acknowledges support from the "Giorgio Mortara" scholarship by the Bank of Italy. NH acknowledges support from the NSF IFML 2019844 and the NSF AI Institute for Foundations of Machine Learning.

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

# Supplement to "Bayesian Nonparametrics Meets Data-Driven Distributionally Robust Optimization"

This Supplement to "Bayesian Nonparametrics Meets Data-Driven Distributionally Robust Optimization" is organized as follows. In Appendix A, we collect the proofs of all results presented in the main text. In Appendix B, we provide further background on Dirichlet process representations and related posterior simulation algorithms. In Appendix C, we describe in detail gradient based optimization of our criterion and the experiments presented in the paper. Finally, in Appendix D we provide details on our computational infrastructure.

## A    Technical Proofs and Further Results

**Proof of Proposition 3.1.**    Let $\rightsquigarrow$ denote weak convergence of probability measures. By Corollary 4.17 in [17], $Q_{\boldsymbol{\xi}^n} \rightsquigarrow \delta_{p_*}$ almost surely. That is,

$$\lim_{n\to\infty} \int_{\mathscr{P}_\Xi} f(p) Q_{\boldsymbol{\xi}^n}(\mathrm{d}p) = f(p_*)$$

almost surely for any bounded and continuous $f : \mathscr{P}_\Xi \to \mathbb{R}$. Thus, we are left to prove that $p \mapsto \phi(\mathcal{R}_p(\theta))$ is bounded and continuous for all $\theta \in \Theta$. It is bounded because $\phi$ is continuous on the compact interval $[0, K]$, and it is continuous because $p \mapsto \mathcal{R}_p(\theta)$ is continuous (by the definition of topology of weak convergence and because $\xi \mapsto h(\theta, \xi)$ is bounded and continuous) and $\phi$ is continuous.

**Proof of Lemma 3.2.**    First note that, by the stated assumptions, it follows from Taylor's theorem that

$$\phi(\mathcal{R}_p(\theta)) = \phi(\mathcal{R}_{p_*}(\theta)) + \phi'(\mathcal{R}_{p_*}(\theta))[\mathcal{R}_p - \mathcal{R}_{p_*}] + \frac{\phi''(c_{p,\theta})}{2}[\mathcal{R}_p - \mathcal{R}_{p_*}]^2$$

for all $p \in \mathscr{P}_\Xi$ and $\theta \in \Theta$ and for some $c_{p,\theta} \in [0, K]$. Then

$$\sup_{\theta\in\Theta} |V_{\boldsymbol{\xi}^n}(\theta) - \phi(\mathcal{R}_{p_*}(\theta))| = \sup_{\theta\in\Theta} \left| \phi'(\mathcal{R}_{p_*}(\theta)) \int_{\mathscr{P}_\Xi} [\mathcal{R}_p(\theta) - \mathcal{R}_{p_*}(\theta)] Q_{\boldsymbol{\xi}^n}(\mathrm{d}p) \right.$$

$$\left. + \int_{\mathscr{P}_\Xi} \frac{\phi''(c_{p,\theta})}{2}[\mathcal{R}_p(\theta) - \mathcal{R}_{p_*}(\theta)]^2 Q_{\boldsymbol{\xi}^n}(\mathrm{d}p) \right|$$

$$\leq M_\phi \sup_{\theta\in\Theta} \left| \frac{\alpha}{\alpha+n} \mathcal{R}_{p_0}(\theta) + \frac{n}{\alpha+n} \mathcal{R}_{p_{\boldsymbol{\xi}^n}}(\theta) - \mathcal{R}_{p_*}(\theta) \right|$$

$$+ M_\phi \sup_{\theta\in\Theta} \left( \int_{\mathscr{P}_\Xi} \frac{[\mathcal{R}_p(\theta) - \mathcal{R}_{p_*}(\theta)]^2}{2} Q_{\boldsymbol{\xi}^n}(\mathrm{d}p) \right) \sup_{t\in(0,K)} \gamma_\phi(t)$$

$$\leq M_\phi \left[ \frac{n}{\alpha+n} \sup_{\theta\in\Theta} |\mathcal{R}_{p_{\boldsymbol{\xi}^n}}(\theta) - \mathcal{R}_{p_*}(\theta)| + \frac{\alpha}{\alpha+n} K + \frac{K^2}{2} \sup_{t\in(0,K)} \gamma_\phi(t) \right].$$

**Proof of Theorem 3.3.**    Notice the following decomposition:

$$\underbrace{\phi(\mathcal{R}_{p_*}(\theta_n)) - \phi(\mathcal{R}_{p_*}(\theta_*))}_{\geq 0} \tag{4}$$

$$= \phi(\mathcal{R}_{p_*}(\theta_n)) - V_{\boldsymbol{\xi}^n}(\theta_n) + \underbrace{V_{\boldsymbol{\xi}^n}(\theta_n) - V_{\boldsymbol{\xi}^n}(\theta_*)}_{\leq 0} + V_{\boldsymbol{\xi}^n}(\theta_*) - \phi(\mathcal{R}_{p_*}(\theta_*))$$

$$\leq 2 \sup_{\theta\in\Theta} |V_{\boldsymbol{\xi}^n}(\theta) - \phi(\mathcal{R}_{p_*}(\theta))|.$$

Then, Lemma 3.2 implies that, for all $\delta > 0$,

$$\mathbb{P}[\phi(\mathcal{R}_{p_*}(\theta_n)) - \phi(\mathcal{R}_{p_*}(\theta_*)) \leq \delta]$$

$$\geq \mathbb{P}\left[\sup_{\theta \in \Theta} |V_{\boldsymbol{\xi}^n}(\theta) - \phi(\mathcal{R}_{p_*}(\theta))| \leq \delta/2\right]$$

$$\geq \mathbb{P}\left[M_\phi\left\{\frac{n}{\alpha + n}\sup_{\theta \in \Theta}|\mathcal{R}_{p_{\boldsymbol{\xi}^n}}(\theta) - \mathcal{R}_{p_*}(\theta)| + \frac{\alpha}{\alpha + n}K + \frac{K^2}{2}\sup_{t \in (0,K)}\gamma_\phi(t)\right\} \leq \delta/2\right]$$

$$= \mathbb{P}\left[\sup_{\theta \in \Theta}\left|\mathcal{R}_{p_{\boldsymbol{\xi}^n}}(\theta) - \mathcal{R}_{p_*}(\theta)\right| \leq \frac{\alpha + n}{n}\left(\frac{\delta}{2M_\phi} - \frac{\alpha}{\alpha + n}K - \frac{K^2}{2}\sup_{t \in (0,K)}\gamma_\phi(t)\right)\right].$$

**Proof of Theorem 3.4.** Since

$$\lim_{n \to \infty}\sup_{\theta \in \Theta}|\mathcal{R}_{p_{\boldsymbol{\xi}^n}}(\theta) - \mathcal{R}_{p_*}(\theta)| = 0$$

almost surely and given assumptions (1) and (2) on $\phi_n$, by Lemma 3.2 we obtain

$$\lim_{n \to \infty}\sup_{\theta \in \Theta}|V_{\boldsymbol{\xi}^n}(\theta) - \phi_n(\mathcal{R}_{p_*}(\theta))| = 0$$

almost surely. Then, by decomposition (4),

$$\lim_{n \to \infty}\phi_n(\mathcal{R}_{p_*}(\theta_n)) - \phi_n(\mathcal{R}_{p_*}(\theta_*)) = 0$$

almost surely and

$$\lim_{n \to \infty}V_{\boldsymbol{\xi}^n}(\theta_n) - V_{\boldsymbol{\xi}^n}(\theta_*) = 0$$

almost surely. As a consequence,

$$\lim_{n \to \infty}|V_{\boldsymbol{\xi}^n}(\theta_n) - \phi_n(\mathcal{R}_{p_*}(\theta_*))|$$

$$\leq \lim_{n \to \infty}\left[|V_{\boldsymbol{\xi}^n}(\theta_n) - V_{\boldsymbol{\xi}^n}(\theta_*)| + |V_{\boldsymbol{\xi}^n}(\theta_*) - \phi_n(\mathcal{R}_{p_*}(\theta_*))|\right]$$

$$\leq \lim_{n \to \infty}\left[|V_{\boldsymbol{\xi}^n}(\theta_n) - V_{\boldsymbol{\xi}^n}(\theta_*)| + \sup_{\theta \in \Theta}|V_{\boldsymbol{\xi}^n}(\theta) - \phi_n(\mathcal{R}_{p_*}(\theta))|\right]$$

$$= 0$$

almost surely. Now recall assumption (3), i.e., the sequence $(\phi_n)_{n \geq 1}$ converges uniformly to the identity map. Then, in light of the previous observations and by noticing that

$$|\mathcal{R}_{p_*}(\theta_n) - \mathcal{R}_{p_*}(\theta_*)| \leq |\mathcal{R}_{p_*}(\theta_n) - \phi_n(\mathcal{R}_{p_*}(\theta_n))| + |\phi_n(\mathcal{R}_{p_*}(\theta_n)) - \phi_n(\mathcal{R}_{p_*}(\theta_*))|$$
$$+ |\phi_n(\mathcal{R}_{p_*}(\theta_*)) - \mathcal{R}_{p_*}(\theta_*)|$$

and

$$|V_{\boldsymbol{\xi}^n}(\theta_n) - \mathcal{R}_{p_*}(\theta_*)| \leq |V_{\boldsymbol{\xi}^n}(\theta_n) - \phi_n(\mathcal{R}_{p_*}(\theta_*))| + |\phi_n(\mathcal{R}_{p_*}(\theta_*)) - \mathcal{R}_{p_*}(\theta_*)|,$$

the two desired almost sure limits follow:

$$\lim_{n \to \infty}\mathcal{R}_{p_*}(\theta_n) = \mathcal{R}_{p_*}(\theta_*), \qquad \lim_{n \to \infty}V_{\boldsymbol{\xi}^n}(\theta_n) = \mathcal{R}_{p_*}(\theta_*).$$

**Proof of Theorem 3.6.** We have

$$\mathcal{R}_{p_*}(\theta_*) \leq \mathcal{R}_{p_*}(\bar{\theta}) = \mathbb{E}_{\xi \sim p_*}\lim_{n' \to \infty}h(\theta_n, \xi) = \lim_{n \to \infty}\mathcal{R}_{p_*}(\theta_n) = \mathcal{R}_{p_*}(\theta_*)$$

almost surely, where the first equality follows from the continuity of $\theta \mapsto h(\theta, \xi)$ and the second one from the Dominated Convergence Theorem. Then, $\mathcal{R}_{p_*}(\bar{\theta}) = \mathcal{R}_{p_*}(\theta_*)$ almost surely, proving the result.

To Prove Lemma 4.3, we introduce two other Lemmas. After proving those, Lemma 4.3 follows immediately.

**Lemma A.1.** *For all $T, N \geq 1$,*

$$\sup_{\theta \in \Theta} |\hat{V}_{\boldsymbol{\xi}^n}(\theta, T, N) - V_{\boldsymbol{\xi}^n}(\theta)| \leq M_\phi K \left( \frac{\alpha + n}{\alpha + n + 1} \right)^T + \sup_{\theta \in \Theta} \left| \hat{V}_{\boldsymbol{\xi}^n}(\theta, T, N) - \mathbb{E}[\hat{V}_{\boldsymbol{\xi}^n}(\theta, T, 1)] \right|. \quad (5)$$

*Proof.* We have

$$\sup_{\theta \in \Theta} |\hat{V}_{\boldsymbol{\xi}^n}(\theta, T, N) - V_{\boldsymbol{\xi}^n}(\theta)|$$

$$\leq \sup_{\theta \in \Theta} \left| \hat{V}_{\boldsymbol{\xi}^n}(\theta, T, N) - \mathbb{E}[\hat{V}_{\boldsymbol{\xi}^n}(\theta, T, 1)] \right| + \sup_{\theta \in \Theta} \left| \mathbb{E}[\hat{V}_{\boldsymbol{\xi}^n}(\theta, T, 1)] - V_{\boldsymbol{\xi}^n}(\theta) \right|. \quad (6)$$

Note that

$$\mathbb{E}[\hat{V}_{\boldsymbol{\xi}^n}(\theta, T, 1)] = \mathbb{E}_{p_1, \xi_1, \ldots, p_T, \xi_T} \left[ \phi \left( \sum_{j=1}^T p_j h(\theta, \xi_j) + p_0 h(\theta, \xi_0) \right) \right]$$

$$= \mathbb{E}_{\sum_{j \geq 1} p_j \delta_{\xi_j} \sim Q_{\boldsymbol{\xi}^n}} \left[ \phi \left( \sum_{j=1}^T p_j h(\theta, \xi_j) + p_0 h(\theta, \xi_0) \right) \right]$$

$$= \mathbb{E}_{\sum_{j \geq 1} p_j \delta_{\xi_j} \sim Q_{\boldsymbol{\xi}^n}} \left[ \phi \left( \sum_{j=1}^\infty p_j h(\theta, \xi_j) + p_0 h(\theta, \xi_0) - \sum_{j=T+1}^\infty p_j h(\theta, \xi_j) \right) \right]$$

$$= V_{\boldsymbol{\xi}^n}(\theta) + \mathbb{E}_{\sum_{j \geq 1} p_j \delta_{\xi_j} \sim Q_{\boldsymbol{\xi}^n}} \left[ \phi' \left( c_{\theta, \sum_{j \geq 1} p_j \delta_{\xi_j}} \right) p_0 \left\{ h(\theta, \xi_0) - \sum_{j=T+1}^\infty \frac{p_j}{p_0} h(\theta, \xi_j) \right\} \right],$$

where the last equality follows from the mean value theorem applied to endpoints $\sum_{j=1}^\infty p_j h(\theta, \xi_j)$ and $\sum_{j=1}^\infty p_j h(\theta, \xi_j) + p_0 h(\theta, \xi_0) - \sum_{j=T+1}^\infty p_j h(\theta, \xi_j)$. Then the second term in (6) is bounded by

$$M_\phi K \mathbb{E}_{\sum_{j \geq 1} p_j \delta_{\xi_j} \sim Q_{\boldsymbol{\xi}^n}} [p_0] = M_\phi K \mathbb{E} \left[ \prod_{k=1}^T (1 - B_k) \right] = M_\phi K \left( \frac{\alpha + n}{\alpha + n + 1} \right)^T.$$

$\square$

The second term on the left-hand side of Equation(5) is instead of the form

$$\sup_{g \in \mathscr{G}} \left| \frac{1}{N} \sum_{i=1}^N g(X_i) - \mathbb{E}[g(X_1)] \right|, \quad (7)$$

where $X_i = \sum_{j=0}^T p_{ij} h(\theta, \xi_{ij})$ are iid random variables whose distribution is determined by the truncated stick-breaking procedure.

The aim of the next Lemma is to provide sufficient conditions for finite sample bounds and asymptotic convergence to 0 of the term in (7). Specifically, we impose complexity constraints on the function class $\mathscr{H} := \{\xi \mapsto h(\theta, \xi) : \theta \in \Theta\}$ which allow us to obtain appropriate conditions on the derived class

$$\mathscr{F} := \left\{ (p_j, \xi_j)_{j=0}^T \mapsto \phi \left( \sum_{j=1}^T p_j h(\theta, \xi_j) + p_0 h(\theta, \xi_0) \right) : \theta \in \Theta \right\},$$

ensuring the asymptotic and non-asymptotic results we seek.

**Lemma A.2.** *Assume $\Theta$ is a bounded subset of $\mathbb{R}^d$ and $\theta \mapsto h(\theta, \xi)$ is $c(\xi)$-Lipschitz continuous for all $\xi \in \Xi$. Then, for all $T, N \geq 1$ and $\varepsilon > 0$,*

$$\sup_{\theta \in \Theta} \left| \hat{V}_{\boldsymbol{\xi}^n}(\theta, T, N) - \mathbb{E}[\hat{V}_{\boldsymbol{\xi}^n}(\theta, T, 1)] \right| \leq \varepsilon$$

*with probability at least*

$$1 - 2 \left( \frac{32 M_\phi C_T \text{diam}(\Theta) \sqrt{d}}{\varepsilon} \right)^d \left[ \exp \left\{ -\frac{3N\varepsilon^2}{4\phi(K)(6\phi(K) + \varepsilon)} \right\} + \exp \left\{ -\frac{3N\varepsilon}{40\phi(K)} \right\} \right].$$

*for some constant $C_T > 0$.*

*Proof.* By the Lipschitz continuity assumption on $h(\theta, \xi)$, we obtain that, for all $(p_j, \xi_j)_{j=0}^T$, $\theta \mapsto \phi\left(\sum_{j=1}^T p_j h(\theta, \xi_j) + p_0 h(\theta, \xi_0)\right)$ is $M_\phi \tilde{c}\left((p_j, \xi_j)_{j=0}^T\right)$-Lipschitz continuous, with

$$\tilde{c}\left((p_j, \xi_j)_{j=0}^T\right) := \sum_{j=0}^T p_j c(\xi_j).$$

Indeed, for all $\theta_1, \theta_2 \in \Theta$,

$$\left| \phi\left(\sum_{j=1}^T p_j h(\theta_1, \xi_j) + p_0 h(\theta_1, \xi_0)\right) - \phi\left(\sum_{j=1}^T p_j h(\theta_2, \xi_j) + p_0 h(\theta_2, \xi_0)\right) \right|$$

$$\leq M_\phi \sum_{j=1}^T p_j |h(\theta_1, \xi_j) - h(\theta_2, \xi_j)| + p_0 |h(\theta_1, \xi_0) - h(\theta_2, \xi_0)|$$

$$\leq M_\phi \tilde{c}\left((p_j, \xi_j)_{j=0}^T\right) \|\theta_1 - \theta_2\|.$$

Therefore, denoting by $P$ the law of the vector $(p_j, \xi_j)_{j=0}^T$ and by $N_{[]}(\varepsilon, \mathscr{F}, \mathcal{L}^1(P))$ the associated $\varepsilon$-bracketing number of the class $\mathscr{F}$, by Lemma 7.88 in [25] we obtain

$$N_{[]}(\varepsilon, \mathscr{F}, \mathcal{L}^1(P)) \leq \left(\frac{4 M_\phi C \mathrm{diam}(\Theta)\sqrt{d}}{\varepsilon}\right)^d,$$

with $C_T := \int \tilde{c}\left((p_j, \xi_j)_{j=0}^T\right) \mathrm{d}P$. Then the result follows by Theorem 7.86 in [25] after noticing that $\sup_{f \in \mathscr{F}} \|f\|_{\mathcal{L}^1(P)} \leq \sup_{f \in \mathscr{F}} \|f\|_\infty \leq \phi(K)$. $\qquad \square$

**Proof of Theorem 4.5.** We have

$$V_{\boldsymbol{\xi}^n}(\theta_n) \leq V_{\boldsymbol{\xi}^n}(\bar{\theta}_n)$$

$$= \mathbb{E}_{p \sim Q_{\boldsymbol{\xi}^n}} \left[ \lim_{N \to \infty} \lim_{T \to \infty} \phi(\mathcal{R}_p(\hat{\theta}_n(T, N))) \right]$$

$$= \lim_{N \to \infty} \lim_{T \to \infty} V_{\boldsymbol{\xi}^n}(\hat{\theta}_n(T, N))$$

$$= V_{\boldsymbol{\xi}^n}(\theta_n)$$

almost surely, where the first two equalities follow from the continuity of $\theta \mapsto h(\theta, \xi)$ and $\phi$ as well as from an iterated application of the Dominated Convergence Theorem (recall that $h(\theta, \xi) \in [0, K]$ by assumption for all $\theta$ and $\xi$). This implies $V_{\boldsymbol{\xi}^n}(\theta_n) = V_{\boldsymbol{\xi}^n}(\bar{\theta}_n)$ almost surely.

In the next result, we will assume $\phi(t) \equiv \phi_\beta(t) = \beta \exp(t/\beta) - \beta$. Moreover, we will emphasize, through superscripts, the dependence of mathematical objects on $\beta$ and the DP concentration parameter $\alpha$. Moreover, if necessary, we make the truncation threshold $T_n$ and the number of MC samples $N_n$ dependent on the sample size $n$.

**Proposition A.3.** *Assume $\Theta$ is an open subset of $\mathbb{R}^d$ and $\lim_{N \to \infty} \lim_{T \to \infty} \sup_{\theta \in \Theta} |\hat{V}_{\boldsymbol{\xi}^n}^{\alpha,\beta}(\theta, T, N) - V_{\boldsymbol{\xi}^n}^{\alpha,\beta}(\theta)| = 0$ almost surely for all $\alpha > 0$ and $\beta > 0$. Moreover, assume that*

1. *$\theta \mapsto h(\xi, \theta)$ is differentiable at $\theta_* \in \arg\min_{\theta \in \Theta} \mathcal{R}_{p_*}(\theta)$ for $p_*$-almost every $\xi \in \Xi$, with gradient $\nabla_{\theta_*}(\xi)$;*

2. *For all $\theta_1$ and $\theta_2$ in a neighborhood of $\theta_*$, there exists a measurable function $\xi \mapsto H(\xi) \in \mathcal{L}_{p_*}^2$ such that $|h(\theta_1, \xi) - h(\theta_2, \xi)| \leq H(\xi) \|\theta_1 - \theta_2\|$;*

3. *$\theta \mapsto \mathcal{R}_{p_*}(\theta)$ admits a second-order Taylor expansion at $\theta_*$, with non-singular symmetric Hessian matrix $V_{\theta_*}$.*

*Then, with probability 1, there exist sequences $(T_n)_{n \geq 1}$, $(N_n)_{n \geq 1}$, $(\beta_n)_{n \geq 1}$, (diverging to $\infty$) and $(\alpha_n)_{n \geq 1}$ (converging to 0), such that*

$$\sqrt{n}\left(\hat{\theta}_n^{\alpha_n, \beta_n}(T_n, N_n) - \theta_*\right) \rightsquigarrow \mathcal{N}(0, V), \qquad V := V_{\theta_*}^{-1} \mathbb{E}_{\xi \sim p_*}\left[\nabla_{\theta_*}(\xi) \nabla_{\theta_*}(\xi)^\top\right] V_{\theta_*}^{-1},$$

*provided $\hat{\theta}_n^{\alpha_n, \beta_n}(T_n, N_n) \xrightarrow{p} \theta_*$ as $n \to \infty$.*

*Proof.* The imposed assumptions match those listed in Theorem 5.23 of [42]. The only condition left to prove is that there exist sequences $(T_n)_{n\geq 1}$, $(N_n)_{n\geq 1}$, $(\beta_n)_{n\geq 1}$, (diverging to $\infty$) and $(\alpha_n)_{n\geq 1}$ (converging to 0), such that $\mathcal{R}_{p_{\xi^n}}(\hat{\theta}_n^{\alpha_n,\beta_n}(T_n,N_n)) - \inf_{\theta\in\Theta}\mathcal{R}_{p_{\xi^n}}(\theta) \leq o_p(n^{-1})$. We do so by proving that, for any fixed $n \geq 1$, $\mathcal{R}_{p_{\xi^n}}(\hat{\theta}_n^{\alpha,\beta}(T,N)) \to \inf_{\theta\in\Theta}\mathcal{R}_{p_{\xi^n}}(\theta)$ almost surely as $\alpha \to 0$ and $\beta, T, N \to \infty$; this implies that, for all $n$, with probability 1 there exist $\alpha_n, \beta_n, T_n$ and $N_n$ such that $\mathcal{R}_{p_{\xi^n}}(\hat{\theta}_n^{\alpha_n,\beta_n}(T_n,N_n)) - \inf_{\theta\in\Theta}\mathcal{R}_{p_{\xi^n}}(\theta) \leq \varepsilon_n$ for any $\varepsilon_n > 0$. Moreover, it is easy to see that the result is implied by $\sup_{\theta\in\Theta}|\hat{V}_{\xi^n}^{\alpha,\beta}(\theta,T,N) - \mathcal{R}_{p_{\xi^n}}(\theta)| \to 0$, so we prove the latter. We have

$$\sup_{\theta\in\Theta}|\hat{V}_{\xi^n}^{\alpha,\beta}(\theta,T,N) - \mathcal{R}_{p_{\xi^n}}(\theta)|$$
$$\leq \sup_{\theta\in\Theta}|\hat{V}_{\xi^n}^{\alpha,\beta}(\theta,T,N) - V_{\xi^n}^{\alpha,\beta}(\theta)| + \sup_{\theta\in\Theta}|V_{\xi^n}^{\alpha,\beta}(\theta) - \mathcal{R}_{p_{\xi^n}}(\theta)|,$$

where the first term converges to 0 almost surely by assumption. Using a second-order Taylor expansion of $\phi_\beta(\mathcal{R}_p(\theta))$ around $\mathcal{R}_{p_{\xi^n}}(\theta)$, the second term, instead, satisfies

$$\sup_{\theta\in\Theta}|V_{\xi^n}^{\alpha,\beta}(\theta) - \mathcal{R}_{p_{\xi^n}}(\theta)| = \sup_{\theta\in\Theta}\left|\int_{\mathscr{P}_\Xi}\phi_\beta(\mathcal{R}_p(\theta))Q_{\xi^n}^\alpha(\mathrm{d}p) - \mathcal{R}_{p_{\xi^n}}(\theta)\right|$$
$$\leq \sup_{\theta\in\Theta}\left|\phi_\beta'(\mathcal{R}_{p_{\xi^n}}(\theta))\left(\int_{\mathscr{P}_\Xi}\mathcal{R}_p(\theta)Q_{\xi^n}^\alpha(\mathrm{d}p) - \mathcal{R}_{p_{\xi^n}}(\theta)\right)\right| + \frac{K^2}{2}\sup_{t\in[0,K]}\phi_\beta''(t)$$
$$\leq \underbrace{\sup_{t\in[0,K]}\phi_\beta'(t)}_{\to 1}\underbrace{\sup_{\theta\in\Theta}\left|\frac{n}{\alpha+n}\mathcal{R}_{p_{\xi^n}}(\theta) + \frac{\alpha}{\alpha+n}\mathcal{R}_{p_0}(\theta) - \mathcal{R}_{p_{\xi^n}}(\theta)\right|}_{\to 0} + \underbrace{\frac{K^2}{2}\sup_{t\in[0,K]}\phi_\beta''(t)}_{\to 0} \to 0,$$

as $\alpha \to 0$ and $\beta \to \infty$. $\qquad\square$

*Remark* A.4. Theorems 3.6 and 4.5 ensure that (a) for all $n \geq 1$, provided $\hat{\theta}_n(T,N)$ converges almost surely to some $\theta_n \in \Theta$, the latter is a minimizer of $V_{\xi^n}(\theta)$; and (b) if the above sequence $(\theta_n)_{n\geq 1}$ converges almost surely to some $\theta_* \in \Theta$, the latter is a minimizer of $\mathcal{R}_{p_*}(\theta)$. Notice that the assumptions required for these results are consistent with the ones of Proposition A.3, so they can be used to justify the condition $\hat{\theta}_n^{\alpha_n,\beta_n}(T_n,N_n) \xrightarrow{p} \theta_*$. For instance, if one assumes almost sure uniqueness of minimizers and almost sure convergence of the above defined sequences, $\hat{\theta}_n^{\alpha_n,\beta_n}(T_n,N_n) \xrightarrow{p} \theta_*$ can be guaranteed leveraging the preceding results.

**Stochastic Gradient Descent Convergence Analysis.** The following results refer to material presented in Appendix C below. For ease of exposition, we fix $B = 1$ and denote by $\mathbb{E}_t$ the expectation operator conditional on the realization of the random index draws $m^1,\ldots,m^t \overset{\text{iid}}{\sim}$ Uniform$(\{1,\ldots,M\})$.

**Proposition A.5.** *Assume that $V$ is convex and that $(\theta_t)_{t\geq 1}$ follows Equation (9)) for some starting value $\theta^0 \in \Theta$ and $B = 1$. Moreover, assume that, for all $\bar{\theta} \in \Theta$,*

$$M^{-1}\sum_{m=1}^M \|\ell_m\phi'(H_m(\theta))\nabla_\theta h(\theta,\xi_m)\|^2 \leq \sigma_\nu^2.$$

*Then*

$$\mathbb{E}_{T-1}[V(\tilde{\theta}^T)] - V(\theta^*) \leq \frac{\|\theta^0 - \theta^*\|^2 + \sigma_\nu^2\sum_{t=0}^T \eta_t^2}{2\sum_{t=0}^T \eta_t},$$

*where $\theta^* \in \arg\min_{\theta\in\Theta} V(\theta)$, $\tilde{\theta}^T := \sum_{t=0}^T \nu_t\theta^t$, and $\nu_t := \frac{\eta_t}{\sum_{t'=0}^T \eta_{t'}}$.*

**Proof of Proposition A.5.** Fix $t = 1,\ldots,T$. We have,

$$\|\theta^{t+1} - \theta^*\|^2 = \|\theta^t - \eta_t\ell_{m^t}\phi'(H_{m^t}(\theta^t))\nabla_\theta h(\theta^t,\xi_{m^t}) - \theta^*\|^2$$
$$= \|\theta^t - \theta^*\|^2 + \eta_t^2\|\ell_{m^t}\phi'(H_{m^t}(\theta^t))\nabla_\theta h(\theta^t,\xi_{m^t})\|^2$$
$$- 2\eta_t(\theta^t - \theta^*)^\top\ell_{m^t}\phi'(H_{m^t}(\theta^t))\nabla_\theta h(\theta^t,\xi_{m^t}).$$

Applying the law of total expectation and the fact that $\ell_{m^t}\phi'(H_{m^t}(\theta^t))\nabla_\theta h(\theta^t, \xi_{m^t})$ is unbiased for $\nabla_\theta V(\theta^t)$,

$$\mathbb{E}_t[(\theta^t - \theta^*)^\top \ell_{m^t}\phi'(H_{m^t}(\theta^t))\nabla_\theta h(\theta^t, \xi_{m^t})] = \mathbb{E}_t[\mathbb{E}_{t-1}[(\theta^t - \theta^*)^\top \ell_{m^t}\phi'(H_{m^t}(\theta^t))\nabla_\theta h(\theta^t, \xi_{m^t})]]$$
$$= \mathbb{E}_{t-1}[(\theta^t - \theta^*)^\top \nabla_\theta V(\theta^t)].$$

Hence

$$2\eta_t \mathbb{E}_{t-1}[(\theta^t - \theta^*)^\top \nabla_\theta V(\theta^t)] = \mathbb{E}_{t-1}[\|\theta^t - \theta^*\|^2] - \mathbb{E}_t[\|\theta^{t+1} - \theta^*\|^2]$$
$$+ \eta_t^2 \mathbb{E}_t[\|\ell_{m^t}\phi'(H_{m^t}(\theta^t))\nabla_\theta h(\theta^t, \xi_{m^t})\|^2]$$
$$\leq \mathbb{E}_{t-1}[\|\theta^t - \theta^*\|^2] - \underbrace{\mathbb{E}_t[\|\theta^{t+1} - \theta^*\|^2]}_{\geq 0} + \sigma_\nu^2.$$

Summing over $t = 0, \ldots, T$ and since

$$\mathbb{E}_{t-1}[(\theta^t - \theta^*)^\top \nabla_\theta V(\theta^t)] \geq \mathbb{E}_{t-1}[V(\theta^t) - V(\theta^*)]$$

because $V$ is convex, we have

$$2\sum_{t=0}^T \eta_t \mathbb{E}_{t-1}[V(\theta^t) - V(\theta^*)] \leq \|\theta^0 - \theta^*\|^2 + \sigma_\nu^2 \sum_{t=0}^T \eta_t^2.$$

Dividing both sides by $\sum_{t=0}^T \eta_t$ and exploiting (i) the linearity of the expectation operator, (ii) the convexity of the weights $(\nu_t)_{t=0}^T$, and (iii) the convexity of $V$, the result follows.

**Proof of Proposition 2.1.** As for case 1, given the assumed form of $p_0$ and the criterion representation (2), we are left to establish an expression for $\mathbb{E}_{\xi \sim p_0}[h(\theta, \xi)] = \mathbb{E}_{(y,x)\sim\mathcal{N}(0,I)}[(y - \theta^\top x)^2]$. Notice that $-\theta_j x_j \overset{\text{id}}{\sim} \mathcal{N}(0, \theta_j^2)$, independendently of $y \sim \mathcal{N}(0, 1)$, so that $y - \theta^\top x \sim \mathcal{N}(0, 1 + \|\theta\|_2^2)$. Therefore, $\mathbb{E}_{\xi \sim p_0}[h(\theta, \xi)] = \mathbb{V}[y - \theta^\top x] = 1 + \|\theta\|_2^2$, which is easily seen to complete the proof. Finally, the proof for the LASSO case is completely analogous to the Ridge one and is therfore omitted.

# B    Further Background on the Dirichlet Process and Approximation Algorithms

Since its definition by [13] based on the family of finite-dimensional Dirichlet distributions (as sketched in Section 2), the Dirichlet process has been characterized (and thus generalized) in a number of useful ways. For instance, the DP can be derived as a neutral to the right process [14], a normalized completely random measure [13, 22, 26, 36], a Gibbs-type prior [20, 9], a Pitman-Yor Process [33, 32], and a species sampling model [34]. In what follows, we review two other constructions of the DP which were at the basis of the approximate versions of the robust criterion $V_{\xi^n}$ proposed in Section 4.

**Stick-Breaking Construction of the Dirichlet Process.** [39] proved that Ferguson's 1973 Dirichlet process enjoys the following "stick-breaking" representation

$$p \sim \text{DP}(\alpha, P) \implies p \overset{\text{d}}{=} \sum_{j=1}^\infty p_j \delta_{x_j},$$

where

$$x_j \overset{\text{iid}}{\sim} P, \quad j = 1, 2, \ldots,$$
$$p_1 = B_1,$$
$$p_j = B_j \prod_{i=1}^{j-1} B_i, \quad j = 2, 3, \ldots,$$
$$B_j \overset{\text{iid}}{\sim} \text{Beta}(1, \alpha), \quad j = 1, 2, \ldots$$

---

**Algorithm 1** SBMC Approximation

---

**Input:** Data $\boldsymbol{\xi}^n$, model parameters, number of MC samples $N$, truncation step $T$
**for** $i = 1$ **to** $N$ **do**
  Set $\prod_{k=1}^{0}(1 - B_k) \equiv 1$
  **for** $j = 1$ **to** $T$ **do**
    Draw $\xi_{ij} \sim \frac{\alpha}{\alpha + n} p_0 + \frac{n}{\alpha + n} p_{\boldsymbol{\xi}^n}$
    Draw $B_{ij} \sim \text{Beta}(1, \alpha + n)$
    Set $p_{ij} = B_j \prod_{k=1}^{j-1}(1 - B_k)$
  **end for**
  Draw $\xi_{i0} \sim \frac{\alpha}{\alpha + n} p_0 + \frac{n}{\alpha + n} p_{\boldsymbol{\xi}^n}$
  Set $p_{i0} = \prod_{k=1}^{T}(1 - B_k)$
**end for**
**Return:** $N^{-1} \sum_{i=1}^{N} \phi\big( \sum_{j=0}^{T} p_{ij} h(\theta, \xi_{ij}) \big)$

---

The name of the procedure comes from the analogy with breaking a stick of length 1 into two pieces of length $B_1$ and $1 - B_1$, then the second piece into two sub-pieces of length $(1 - B_1)B_2$ and $(1 - B_1)(1 - B_2)$, and so on. In Algorithm 1, then, we simulate $N$ realizations from $Q_{\boldsymbol{\xi}^n}$, truncating the stick-breaking procedure at step $j = T$. The remaining portion of the stick is then allocated to one further atom drawn from the predictive distribution. Then, the intractable integral with respect to the DP posterior is approximated via a Monte Carlo average of the integrals (i.e., weighted sums) with respect to the $N$ simulated measures.

**Multinomial-Dirichlet Construction of the Dirichlet Process and Monte Carlo Algorithms.**
Another finite-dimensional approximation of $p \sim \text{DP}(\alpha, P)$ is $p_T = \sum_{j=1}^{T} p_j \delta_{x_j}$, with $x_j \overset{\text{iid}}{\sim} P$ and $(p_1, \ldots, p_T) \sim \text{Dirichlet}(T; \alpha/T, \ldots, \alpha/T)$. As $T \to \infty$, $p_T$ approaches $p$ [see Theorem 4.19 in 17]. Hence, one can approximate $V_{\boldsymbol{\xi}^n}(\theta)$ as in Algorithm 2, where the concentration parameter is $\alpha + n$ and the centering distribution coincides with the predictive.

---

**Algorithm 2** Multinomial-Dirichlet Monte Carlo (MDMC) Approximation

---

**Input:** Data $\boldsymbol{\xi}^n$, model parameters, number of MC samples $N$, approximation threshold $T$
**for** $i = 1$ **to** $N$ **do**
  Initialize $\boldsymbol{w}_i \in \mathbb{R}^T, \boldsymbol{\xi}_i \in \Xi^T$
  **for** $j = 1$ **to** $T$ **do**
    Update $\boldsymbol{w}_i(j) \sim \text{Gamma}\big( \frac{\alpha + n}{T}, 1 \big)$
    Update $\boldsymbol{\xi}_i(j) \sim \frac{\alpha}{\alpha + n} p_0 + \frac{n}{\alpha + n} p_{\boldsymbol{\xi}^n}$
  **end for**
  Normalize $\boldsymbol{w}_i = \frac{\boldsymbol{w}_i}{\sum_{j=1}^{n} \boldsymbol{w}_i(j)}$
**end for**
**Return:** $N^{-1} \sum_{i=1}^{N} \phi\big( \boldsymbol{w}_i^\top h(\theta, \boldsymbol{\xi}_i) \big)$

---

When $\alpha$ is negligible compared to the sample size $n$, one can simplify posterior simulation by setting $\alpha = 0$. Thus, one obtains a $\text{DP}(n, p_{\boldsymbol{\xi}^n})$ posterior. This distribution enjoys a useful representation as follows: $p \sim \text{DP}(n, p_{\boldsymbol{\xi}^n}) \implies p \overset{\text{d}}{=} \sum_{i=1}^{n} p_i \xi_i$, with $p_i \sim \text{Dirichlet}(n; 1, \ldots, 1)$ [see 17, Section 4.7]. Due to its similarity to the usual bootstrap procedure [12], this distribution is known as the "Bayesian bootstrap". Algorithm 3 implements the Bayesian bootstrap to approximate the criterion $V_{\boldsymbol{\xi}^n}(\theta)$. In practice, however, we do not recommend resorting to the Bayesian bootstrap approximation, since $\text{DP}(n, p_{\boldsymbol{\xi}^n})$ assigns probability 1 to the set of distributions with strictly positive support on $\boldsymbol{\xi}^n$. This goes against the prescription that, as the finite sample $\boldsymbol{\xi}^n$ provides only partial information on the true underlying distribution, the statistical DM should be willing to consider a wider set of distributions other than the ones supported at the sample realizations.

---
**Algorithm 3** Bayesian Bootstrap Monte Carlo (BBMC) Approximation
---
   **Input:** Data $\boldsymbol{\xi}^n$, model parameters, number of MC samples $N$
   **for** $i = 1$ **to** $N$ **do**
      Initialize $\boldsymbol{w}_i \in \mathbb{R}^n$
      **for** $j = 1$ **to** $n$ **do**
         Update $\boldsymbol{w}_i(j) \sim \text{Gamma}(1, 1)$
      **end for**
      Normalize $\boldsymbol{w}_i = \frac{\boldsymbol{w}_i}{\sum_{j=1}^n \boldsymbol{w}_i(j)}$
   **end for**
   **Return:** $N^{-1} \sum_{i=1}^N \phi\big(\boldsymbol{w}_i^\top h(\theta, \boldsymbol{\xi}^n)\big)$
---

## C   Numerical Optimization and Experiment Details

In this Section, we first describe the SGD algorithm used in practice for our experiments. Then, we describe in full detail the experiments presented in the paper.

**Gradient-Based Optimization.**   Whether we resort to the SBMC or the MDMC approximation of $V_{\boldsymbol{\xi}^n}(\theta)$, we are faced with the task of minimizing a criterion of the form

$$V(\theta) = \frac{1}{N} \sum_{i=1}^N \phi \left( \sum_{j=1}^T p_{ij} h(\theta, \xi_{ij}) \right).$$

The smoothness and convexity of $\phi$ make it appealing to minimize the criterion via gradient-based convex optimization techniques. Indeed, it is enough to assume that $\theta \mapsto h(\theta, \xi)$ is convex and differentiable (a standard assumption met in many applications of interest) to easily yield the same properties for $V$.

In light of this, assuming that $\theta \mapsto h(\theta, \xi)$ is differentiable at every $\xi_{ij}$ and denoting $H_i(\theta) := \sum_{j=1}^T p_{ij} h(\theta, \xi_{ij})$, the gradient of $V$ is

$$\nabla_\theta V(\theta) = \frac{1}{N} \sum_{i=1}^N \phi'(H_i(\theta)) \nabla_\theta H_i(\theta) \equiv \frac{1}{M} \sum_{m=1}^M \ell_m \phi'(H_m(\theta)) \nabla_\theta h(\theta, \xi_m), \tag{8}$$

where $\ell_m \equiv T p_m$ and the $m$-indexing is just a recoding of the indices (with a slight abuse of notation and $M \equiv N \cdot T$). That is, the gradient of $V(\theta)$ can be written as the average of $M$ terms. Thus, to minimize $V(\theta)$ we propose a mini-batch Stochastic Gradient Descent algorithm which, at each iteration $t$, updates the parameter vector as follows:

$$\theta^{t+1} = \theta^t - \eta_t \frac{1}{B} \sum_{m_b=1}^B \ell_{m_b} \phi'(H_{m_b}(\theta^t)) \nabla_\theta h(\theta^t, \xi_{m_b}), \tag{9}$$

for a step-size $\eta_t > 0$ and a random subset (mini-batch) of size $B$ from the indices $\{1, \ldots, M\}$. Under standard regularity assumptions [16], in Proposition A.5 (Appendix A) we prove convergence of the algorithm at usual rates for convex problems.

*Remark* C.1. Expression (8) provides some insight on how, in practice, distributional robustness is enforced. Notice that $H_i(\theta) = \mathcal{R}_{p_i}(\theta)$ is the expected risk computed according to $p_i$, an approximate realization from $Q_{\boldsymbol{\xi}^n}$. Thus, in the computation of the overall gradient $\nabla_\theta V(\theta)$, the gradients associated to the $p_i$'s that generate higher expected risks receive more weight (being $\phi$ convex, $\phi'$ is increasing). These weights, then, are reflected into which gradients, in the mini-batch SGD algorithm, are given more leverage in updating the parameter vector. Thus, the procedure can be thought of as implementing a "soft worst-case scenario" scheme, whereby distributions in the posterior support are weighted (in terms of gradient influence) more the worse they do in terms of expected risk.

**Mini-Batch Stochastic Gradient Descent Algorithm.**   For practical optimization, we apply a modification to the SGD algorithm provided in Equation (9), which helps to reduce the computational

burden of the procedure. Indeed, recall the formula of the gradient of the criterion $V$ that we need to optimize:

$$\nabla_\theta V(\theta) = \frac{1}{N} \sum_{i=1}^{N} \phi'(H_i(\theta)) \nabla_\theta H_i(\theta)$$

$$\equiv \frac{1}{M} \sum_{m=1}^{M} \ell_m \phi'(H_m(\theta)) \nabla_\theta h(\theta, \xi_m).$$

Clearly, then, implementing the baseline SGD algorithm requires, at each iteration, the evaluation of multiple $H_m(\theta^t)$ terms, each consisting of $T$ evaluations of the loss function $h$. To avoid this, at each iteration we instead sub-sample one index $i = 1, \ldots, N$ and update the parameter vector according to the associated gradient $\phi'(H_i(\theta^t)) \nabla_\theta H_i(\theta^t)$. The latter is still an unbiased estimator of the overall gradient of $V(\theta^t)$, but it requires only $T$ evaluations of $h$ (plus those of $T$ gradients of $h$, similarly to the baseline algorithm). Finally, to exploit the whole data efficiently, we sub-sample without replacement and perform multiple passes over the $N$ MC samples. Algorithm 4 summarizes the procedure.

---

**Algorithm 4** Modified Stochastic Gradient Descent Algorithm

---

**Input:** Approximate criterion parameters $\{(p_{ij}, \xi_{ij}) : i = 1, \ldots, N, j = 1, \ldots, T\}$, step size schedule $(\eta_t)_{t \geq 0}$, starting value $\theta^0$, number of passes $P$, iteration tracker $t = 0$
**for** $p = 1$ **to** $P$ **do**
   Initialize $I = \{1, \ldots, N\}$
   **for** $j = 1$ **to** $N$ **do**
      Sample uniformly $i \in I$
      Update $\theta^{t+1} = \theta^t - \eta_t \cdot \phi'\left(\sum_{\ell=1}^{T} p_{i\ell} h(\theta^t, \xi_{i\ell})\right) \cdot \sum_{\ell=1}^{T} p_{i\ell} \nabla_\theta h(\theta^t, \xi_{i\ell})$
      Update $I = I \setminus \{i\}$
      Update $t = t + 1$
   **end for**
**end for**
**Return:** $\theta^{PN+1}$

---

## C.1 High-Dimensional Linear Regression Experiment

**Setting.** In this experiment, we test the performance of our robust criterion in a high-dimensional sparse linear regression task. The high-dimensional and sparse nature of the data-generating process is expected to induce distributional uncertainty, and our method is meant to address this. In this context, we use the quadratic loss function $(\theta, y, x) \mapsto 10^{-3}(y - \theta^\top x)^2$, where the $10^{-3}$ factor serves to stabilize numerical values in the optimization process. Notice that, by the form of the ambiguity-neutral criterion (2), the multiplicative factor on the loss function does not change the equivalence with Ridge.

**Data-Generating Process.** The data for the experiment are generated iid across simulations (200) and observations ($n = 100$ per simulation) as follows. For each observation $i = 1, \ldots, n$, the $d$-dimensional ($d = 90$) covariate vector follows a multivariate normal distribution with mean 0 and such that (i) each covariate has unitary variance, and (ii) any pair of distinct covariates has covariance 0.3:

$$x_i = \begin{bmatrix} x_{i1} \\ \vdots \\ x_{id} \end{bmatrix} \sim \mathcal{N}(0, \Sigma), \quad \Sigma = \begin{bmatrix} 1 & 0.3 & \cdots & 0.3 \\ 0.3 & 1 & \cdots & 0.3 \\ \vdots & \vdots & \ddots & \vdots \\ 0.3 & 0.3 & \cdots & 1 \end{bmatrix} \in \mathbb{R}^{d \times d}.$$

Then, the response has conditional distribution $y_i | x_i \sim \mathcal{N}(a^\top x_i, \sigma^2)$, with $a = (1, 1, 1, 1, 1, 0, \cdots, 0)^\top \in \mathbb{R}^d$ and $\sigma = 0.5$. That is, out of 90 covariates, only the first 5 have a unitary positive marginal effect on $y_i$, and additive Gaussian noise is added to the resulting linear combination. Together with 100 training samples, at each simulation we generate 5000 test samples on which we compute out-of-sample RMSE for the ambiguity-averse, ambiguity-neutral, and OLS procedures.

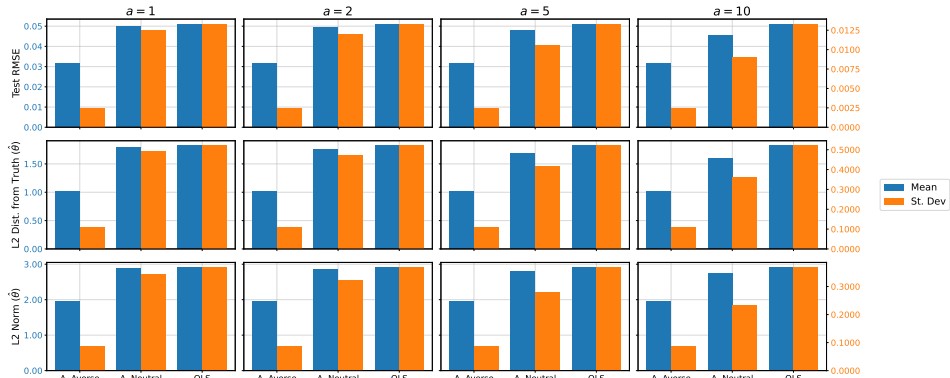

Figure 2: Simulation results for the high-dimensional sparse linear regression experiment. Bars report the mean and standard deviation (across 200 sample simulations) of the test RMSE, $L_2$ distance of estimated coefficient vector $\hat{\theta}$ from the data-generating one, and the $L_2$ norm of $\hat{\theta}$. Results are shown for the ambiguity-averse, ambiguity-neutral, and OLS procedures. Note: The left (blue) axis refers to mean values, the right (orange) axis to standard deviation values.

**Robust Criterion Parameters.** For each simulated sample, we run our robust procedure setting the following parameter values: $\phi(t) = \beta \exp(t/\beta) - \beta$, $\beta \in \{1, \infty\}$, $\alpha = a/n$ for $a \in \{1, 2, 5, 10\}$, and $p_0 = \mathcal{N}(0, I)$, where the $\beta = \infty$ setting corresponds to Ridge regression with regularization parameter $\alpha$ (see Proposition 2.1). Finally, we run 300 Monte Carlo simulations to approximate the criterion, and truncate the Multinomial-Dirichlet approximation at $T = 50$.

**Stochastic Gradient Descent Parameters** We initialize the algorithm at $\theta = (0, \ldots, 0)$ and set the step size at $\eta_t = 50/(100 + \sqrt{t})$. The number of passes over data is set after visual inspection of convergence of the criterion value. The run time per SGD run is less than 1 second on our infrastructure (see Appendix D).

## C.2 Experiment on Gaussian Location Estimation With Outliers

**Setting.** In this experiment, we test the performance of our robust criterion on the task of estimating a univariate Gaussian mean (assuming the variance is known) when the data is corrupted by a few observations coming from a distant distribution. Clearly, this is a situation where a considerable level of distributional uncertainty is warranted. In this setting, the loss function $h(\xi, \theta) = (\xi - \theta)^2$ is simply the negative log-likelihood associated to the normal model. Notice that the $h$ is convex in $\theta$ and, as in the previous experiment, we pre-multiply it by a factor of $10^{-3}$ for numerical stability reasons.

**Data-Generating Process.** The data for the experiment are generated iid across simulations (100) and observations ($n = 13$ per simulation) as follows. For each simulation, 10 iid samples $x_i$ are drawn from a $\mathcal{N}(0, 1)$ distribution (the actual data-generating process we want to learn) and 3 samples are drawn iid from a $\mathcal{N}(0, 5)$ outlier distribution. At each simulation we also generate 5000 test samples from the data-generating process $\mathcal{N}(0, 1)$, on which we compute the out-of-sample average negative log-likelihood for the ambiguity-averse, ambiguity-neutral, and Maximum Likelihood Estimation (MLE) procedures – this will be our measure of out of sample performance (see Figure 3).

**Robust Criterion Parameters.** For each simulated sample, we run our robust procedure setting the following parameter values: $\phi(t) = \beta \exp(t/\beta) - \beta$, $\beta \in \{1, \infty\}$, $\alpha \in \{1, 2, 5, 10\}$, and $p_0 = \mathcal{N}(\mu_0, I)$, where $\mu_0 = (10 \cdot 0 + 3 \cdot 5)/(10 + 3)$ is a weighted average of the data-generating and the outlier means. By the expression of the ambiguity-neutral criterion (2), it is easy to show that the $\beta = \infty$ case leads to the parameter estimate

$$\hat{\theta}_{\boldsymbol{\xi}^n} = \frac{1}{\alpha + n} \sum_{i=1}^{\alpha + n} y_i,$$

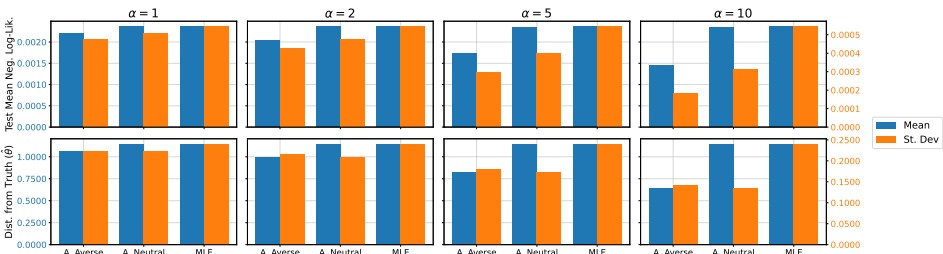

Figure 3: Simulation results from the experiment on Gaussian mean estimation with outliers. Bars report the mean and standard deviation (across 100 sample simulations) of the test mean negative log-likelihood and the absolute value distance of the estimated parameter from 0 (the data-generating value). Results are shown for the ambiguity-averse, ambiguity-neutral, and MLE procedures. Note: The left (blue) axis refers to mean values, the right (orange) axis to standard deviation values.

with $y_i = x_i$ for $i = 1, \ldots, n$ and $y_i = \mu_0$ for $i = n + 1, \ldots, n + \alpha$. That is, the ambiguity-neutral procedure with concentration parameter $\alpha \in \mathbb{N}$ is equivalent to the MLE procedure when the original training sample is enlarged with $\alpha$ additional observations equal to $\mu_0$. Finally, we run 300 Monte Carlo simulations to approximate the criterion, and truncate the Multinomial-Dirichlet approximation at $T = 50$.

**Stochastic Gradient Descent Parameters.** We initialize the algorithm at $\theta = 0$ and set the step size at $\eta_t = 20/(100 + \sqrt{t})$. The number of passes over data is set after visual inspection of convergence of the criterion value. The run time per SGD run is 2 seconds on our infrastructure (see Appendix D).

**Results.** In Figure 3, we present the results of the simulation study. As for the regression experiment, the ambiguity-averse criterion brings improvement, across $\alpha$ values and compared to the ambiguity-neutral and the simple MLE procedures, both in terms of average performance and in terms of the latter's variabiliy (see the first row of the Figure). From the second row of Figure 3, it also emerges that, on average, the ambiguity-averse procedure is more accurate at estimating the location parameter than the two other methods. Compared to the simple MLE procedure, the variability of the estimated parameter is also significantly smaller. Taken together, these results confirm the theoretical expectation that the ambiguity-averse optimization is effective at hedging against the distributional uncertainty arising in the estimation of corrupted data such as the simulated ones.

## C.3 High-Dimensional Logistic Regression Experiment

**Setting.** In this experiment, we test the performance of our robust criterion on a high-dimensional sparse classification task using the framework of logistic regression. As in the linear regression experiment, the high-dimensional and sparse nature of the data-generating process is expected to induce distributional uncertainty, and our method is meant to address this. In this setting, the loss function is $h(\xi, \theta) = \log(1 + \exp(-y \cdot x^\top \theta))$. As in the previous experiment, we pre-multiply it by a factor of $10^{-3}$ for numerical stability reasons.

**Data-Generating Process.** The data for the experiment are generated iid across simulations (200) and observations ($n = 100$ per simulation) as follows. For each observation $i = 1, \ldots, n$, the $d$-dimensional ($d = 90$) covariate vector follows a multivariate normal distribution with mean 0 and such that (i) each covariate has unitary variance, and (ii) any pair of distinct covariates has covariance 0.3:

$$x_i = \begin{bmatrix} x_{i1} \\ \vdots \\ x_{id} \end{bmatrix} \sim \mathcal{N}(0, \Sigma), \quad \Sigma = \begin{bmatrix} 1 & 0.3 & \cdots & 0.3 \\ 0.3 & 1 & \cdots & 0.3 \\ \vdots & \vdots & \ddots & \vdots \\ 0.3 & 0.3 & \cdots & 1 \end{bmatrix} \in \mathbb{R}^{d \times d}.$$

Then, the response has conditional distribution $y_i | x_i \sim \text{Binary}(\{1, -1\}, p_x)$, with $p_x = 1/(1 + \exp(-x^\top a))$ and $a = (1, 1, 1, 1, 1, 0, \cdots, 0)^\top \in \mathbb{R}^d$. That is, out of 90 covariates, only the first 5 have a unitary positive marginal effect on the log-odds. Together with 100 training samples, at each

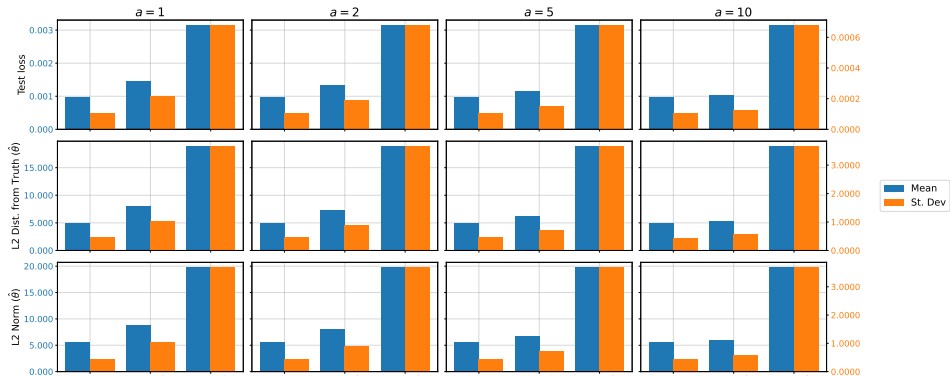

Figure 4: Simulation results for the high-dimensional sparse logistic regression experiment. Bars report the mean and standard deviation (across 200 sample simulations) of the test average loss, $L_2$ distance of estimated coefficient vector $\hat{\theta}$ from the data-generating one, and the $L_2$ norm of $\hat{\theta}$. Results are shown for the ambiguity-averse, $L_2$-regularized, and un-regularized procedures. Note: The left (blue) axis refers to mean values, the right (orange) axis to standard deviation values.

simulation we generate 5000 test samples on which we compute the out-of-sample average loss for the ambiguity-averse, $L_2$-regularized (with regularization parameter $\alpha$, see below), and un-regularized procedures.

**Robust Criterion Parameters.** For each simulated sample, we run our robust procedure setting the following parameter values: $\phi(t) = \beta \exp(t/\beta) - \beta$, $\beta = 1$, $\alpha = a/n$ for $\alpha \in \{1, 2, 5, 10\}$, and $p_0 = \mathrm{Binary}(\{1, -1\}, 0.5) \times \mathcal{N}(0, I)$. Finally, we run 200 Monte Carlo simulations to approximate the criterion, and truncate the Multinomial-Dirichlet approximation at $T = 50$.

**Stochastic Gradient Descent Parameters** We initialize the algorithm at $\theta = (0, \ldots, 0)$ and set the step size at $\eta_t = 1000/(100 + \sqrt{t})$. The number of passes over data is set after visual inspection of convergence of the criterion value. The run time per SGD run is 3 seconds on our infrastructure (see Appendix D).

**Results.** In Figure 3, we present the results of the simulation study. As for the regression experiment, the ambiguity-averse criterion brings improvement, across $\alpha$ values and compared to the $L_2$-regularized and the unregularized procedures, both in terms of average performance and in terms of the latter's variabiliy (see the first row of the Figure). From the second row of Figure 3, it also emerges that, on average, the ambiguity-averse procedure is more accurate and less variable at estimating the true regression coefficient than the two other methods. Also, our method is able to more effectively shrink the norm of the coefficient vector towards 0 (see the third row). Taken together, these results confirm the theoretical expectation that the ambiguity-averse optimization is effective at hedging against the distributional uncertainty arising in high-dimensional classification problems (in this experimental setting, tackled via logistic regression).

### C.4 Pima Indian Diabetes Dataset Experiment

In this experiment, we use logistic regression for classification on the popular Pima Indians Diabetes dataset,[12] collecting data on 768 women belonging to a Native American group that lives in Mexico and Arizona. The data consists of a binary outcome (whether the subject developed diabetes or not) and 8 features related to her physical condition (these features are standardized before running the analysis).

To test our method, we randomly select 300 training observations and leave out the rest for as a test sample. Then, we randomly split the training data into 15 folds of size 20 and select, via $k$-fold cross

---

[12]Made available by the National Institute of Diabetes and Digestive and Kidney Diseases and downloaded from `https://www.kaggle.com/datasets/kandij/diabetes-dataset?resource=download`.

validation, the optimal DP concentration parameter $\alpha$ over a wide grid of values. We do the same for the $L_1$-penalty coefficient used to implement regularized logistic regression with the Python library `scikit-learn` [31]. Once the optimal parameters are selected based on out-of-sample risk, we again randomly split the training sample into the same number of folds, and implement our roubust DP method, L1-penalized logistic regression, and unregularized logistic regression on each of the folds.[13] This splitting procedure allows us (i) to test and compare the performance of our method in a setting with scarce data, where distributional uncertainty is most likely present, and (ii) to asses the sampling variability of the implemented procedures. The run time per SGD run is 19 seconds on our infrastructure (see Appendix D).

Table 1 reports the results from the described procedure. Unregularized logistic regression performs quite poorly compared to the other two methods. Instead, the latter yield results in the same orders of magnitude both in terms of average performance and of performance variability, though our DP robust method features almost half of the variability produced by $L_1$-regularized logistic regression.

|  | Unregularized | $L_1$ Regularized | **DP Robust** |
|---|---|---|---|
| Average | 0.0142 | 0.0007 | 0.0006 |
| Standard Deviation | 0.0127 | 6.2253e-05 | 3.9742e-05 |

Table 1: Comparison of average and standard deviation of the out-of-sample performance (out-of-sample expected logistic loss) of the three employed methods for binary classification on the Pima Indian Diabetes dataset.

## C.5 Wine Quality Dataset Experiment

In this experiment, we applied linear regression to the popular UCI Machine Learning Repository Wine Quality dataset [8]. Data consists of 4898 measurements of 11 wines' characteristics and a quality score assigned to each wine. The aim is to predict the latter based on the former (both features and response are standardized before running the analysis). We implement linear regression using our DP-based robust method (with the squared loss function), OLS, and LASSO (the last two methods are implemented using `scikit-learn` [31]).

To test our method, we randomly select 300 training observations and leave out the rest for as a test sample. Then, we randomly split the training data into 10 folds of size 30 and select, via $k$-fold cross validation, the optimal DP concentration parameter $\alpha$ over a wide grid of values. We do the same for the $L_1$-penalty coefficient used to implement LASSO. Once the optimal parameters are selected based on out-of-sample risk, we again randomly split the training sample into the same number of folds, and implement our roubust DP method, LASSO regression, and OLS estimation on each of the folds. This splitting procedure allows us (i) to test and compare the performance of our method in a setting with scarce data, where distributional uncertainty is most likely present, and (ii) to asses the sampling variability of the implemented procedures. The run time per SGD run is 5 seconds on our infrastructure (see Appendix D).

Table 2 reports the results from the described procedure, whose interpretation is very much in line with the results of the previous experiment.

|  | Unregularized | $L_1$ Regularized | **DP Robust** |
|---|---|---|---|
| Average | 0.0014 | 0.0009 | 0.0009 |
| Standard Deviation | 0.0004 | 8.0192e-05 | 6.0076e-05 |

Table 2: Comparison of average and standard deviation of the out-of-sample performance (out-of-sample expected squared loss) of the three employed methods for linear regression on the Wine Quality dataset.

---

[13]All of the implementation details (e.g., parameter values), can be found in our code. This holds for the next two experiments as well.

### C.6 Liver Disorders Dataset Experiment

In this experiment, we applied linear regression to the popular UCI Machine Learning Repository Liver Disorders dataset[15]. Data consists of 345 measurements of 5 blood test results and the number of drinks consumed per day by each subject. The aim is to predict the latter based on the former (both features and response are standardized before running the analysis). We implement linear regression using our DP-based robust method (with the squared loss function), OLS, and LASSO (the last two methods are implemented using `scikit-learn` [31]).

To test our method, we randomly select 200 training observations and leave out the rest for as a test sample. Then, we randomly split the training data into 10 folds of size 20 and select, via $k$-fold cross validation, the optimal DP concentration parameter $\alpha$ over a wide grid of values. We do the same for the $L_1$-penalty coefficient used to implement LASSO. Once the optimal parameters are selected based on out-of-sample risk, we again randomly split the training sample into the same number of folds, and implement our roubust DP method, LASSO regression, and OLS estimation on each of the folds. This splitting procedure allows us (i) to test and compare the performance of our method in a setting with scarce data, where distributional uncertainty is most likely present, and (ii) to asses the sampling variability of the implemented procedures.The run time per SGD run is 15 seconds on our infrastructure (see Appendix D).

Table 3 reports the results from the described procedure, whose interpretation is very much in line with the results of the previous two experiments.

|  | Unregularized | $L_1$ Regularized | **DP Robust** |
|---|---|---|---|
| Average | 0.0012 | 0.0009 | 0.0007 |
| Standard Deviation | 0.0005 | 0.0001 | 6.6597e-05 |

Table 3: Comparison of average and standard deviation of the out-of-sample performance (out-of-sample expected squared loss) of the three employed methods for linear regression on the Liver Disorders dataset.

## D  Computational Infrastructure

All experiments were performed on a desktop with 12th Gen Intel(R) Core(TM) i9-12900H, 2500 Mhz, 14 Core(s), 20 Logical Processor(s) and 32.0 GB RAM.

