# OpenReview forum: "Bayesian Nonparametrics Meets Data-Driven Distributionally Robust Optimization"
_NeurIPS.cc/2024/Conference — NeurIPS 2024 poster_

### Official Review · Reviewer_p57R · 2024-07-04

**Soundness:** 3
**Presentation:** 2
**Contribution:** 3
**Rating:** 6
**Confidence:** 3

**Summary:**

This article proposes a new method for optimisation of risk under uncertainty, using Dirichlet Processes to introduce an extra degree of robustness to modelling uncertainty on the data generating process. The method is generally applicable to many statistical learning problems through the use of loss functions, and further links are made to the economic decision-making literature. The theoretical properties of the methods are explored, and a Monte Carlo approximation is introduced to make tractable the inference for the DP model. Experiments are performed on three simulated experiments, and three real data sets.

**Strengths:**

This article represents a novel contribution to a very general problem, with the use of a neat mathematical representation of a statistical decision maker’s uncertainty in the form of a Dirichlet Process. The relevant theory behind the method is explored thoroughly, and there are several different applications explored. The quality of the scientific writing is fairly high. The links made with the economic decision making literature are interesting, helping elucidate the underlying point about ambiguity aversion.

**Weaknesses:**

The idea of placing a prior on the data generating process instead of the parameters is unusual from the traditional Bayesian perspective. While this is not necessarily wrong, the authors do not do a particularly good job of communicating the implications of constructing a prior in this way, or how existing intuitions that the readership may have can be transferred to this new approach.

The results discussion from line 299 onwards has not been prioritised for space in the manuscript: the authors attempt to concentrate reporting results from six different experiments into a single page, which is not enough exposition for the main text of the article. Many fairly important details are pushed into an Appendix and then covered by asserting the “superior ability of our robust method” in the main text. This is far too simple an analysis for an empirical investigation: I don’t trust any analysis that can be summed up so concisely, let alone six combined, with three being real data sets. Appendix C is reasonably thorough in this respect, although there is a noticeable absence of any rigorous testing of the differences in performance metrics.

There are a few small issues with the English in the article: line 8: “among which Ridge…”, line 97: “pervasive” is a strange choice of vocabulary.

**Questions:**

Line 110: “In our setting, instead of a parametric prior on the regression coefficients, we place a nonparametric one on the joint distribution of the response and covariates” What is lost by not having priors associated with parameters? This is not necessarily a good thing to move away from: specific belief distributions for specific interpretable model components is a good thing. What the implications for estimating the posterior uncertainty on the parameter level, for example in the regression case? What kind of posterior convergence do you expect on the parameters themselves?

What are the more general implications of putting a prior on the distributions of the data instead of the model parameters? This is suggested to be the case in line 111. Is that not what the likelihood is meant to capture? Is this merely sloppy phrasing in line 111  Does the prior on the distribution of the data come into conflict with the model at some point? What is the difference between altering the prior you have assumed here and altering the statistical model?

What is the expected scalability of the methods to larger data sets? None of those used here are really that big, but the use of SGD implies that you have big-N scalability in mind. The wall clock times on your fairly modest desktop system are quite small, so presumably you have some ability to scale to larger data sets if desired.

Is the trained DP object interpretable at all? Assuming that the learned DP object implies some sort of partition of the data-generating process, then is that partition telling us something we didn’t know beforehand? How restrictive is the DP prior on the space of measures that you might expect in reality?

**Limitations:**

While an elegant choice for the problem at hand, the use of a Dirichlet Process is likely to be a limiting factor for future directions of research, both in terms of the computational limits and the possible allowable partitions.

The assertion of the prior in the way performed here, in contrast to over interpretable parameters, possibly limits the likelihood of this method being widely adopted.

The theoretical results are focussed on convergence to the correct expected risk (or transformations thereof), rather than telling us anything about the parameters themselves.

---

> ### Author Rebuttal · Authors · 2024-08-05
>
> We thank the reviewer for the time spent reviewing our work and for the insightful comments, which we will incorporate in the next version of our paper by emphasizing the points brought up in the review.
>
> First, we would like to address the weaknesses pointed out by the reviewer as follows (we defer discussion of the first weakness to the questions section below):
> * We agree with the reviewer that the results of our experiments are analyzed extremely quickly at the end of the paper. Our choice was dictated by the strict page limit and the predominantly methodological nature of our contribution. However, we will make sure to extend the discussion in the next version of the paper. As for lack of uncertainty quantification for performance metrics, we note how every plot and table reporting empirical results includes standard deviations (across sample realization or data folds) on top of means. This reporting style is due to the fact that variability is of direct interest to our analysis (our method stabilizes out-of-sample performance), but it can be readily used to obtain, for instance, standard normal confidence intervals for the corresponding mean values (that is, mean value $\pm$ standard deviation * critical value).
> * We thank the reviewer for pointing out the language issues, which we will address in the next version of the paper.
>
> Second, we would like to address the reviewer’s questions as follows:
> * The reviewer asks for clarifications on the unusual switch between placing priors on parameters versus placing priors on the data generating process (this also echoes the first weakness mentioned by the reviewer). We’d like to point out that our starting point is data-driven optimization, where parameters of interest are learnt by minimizing some empirical approximation of an expected loss function. The Bayesian component of our methodology lies not in how we infer parameters, but in how we choose to approximate the expectation around the loss function: Instead of directly taking an empirical average, we average out the generating process using a DP posterior. Adopting this perspective, then, also allows to introduce ambiguity aversion via the convex transformation $\phi$. Hence, our method is better understood as using _Bayesian ideas_ to improve upon optimization-based procedures. This allows to gain some key robustness properties and analytical tractability, while retaining the computational scalability of optimization-based learning that traditional Bayesian methods often lack due to burdensome inference on the whole posterior. A good example of this is our Proposition 2.1 (also mentioned by the reviewer), which uncovers a new Bayesian interpretation of Ridge and Lasso. In fact, using traditional Bayesian methods, it is well known that Ridge and Lasso are equivalent to maximum-a-posteriori estimators based on Gaussian and Laplacian priors on the linear regression coefficient vector. Our procedure is instead purely based on optimization of the expected loss function, where the generating process is averaged out via a DP posterior with appropriately chosen centering measure. Of course, as for most black-box optimization procedures, one might argue that uncertainty quantification is harder than with traditional Bayesian methods, and this also holds for our method (however, see Proposition A.3 in the Appendix for an asymptotic normality result). This comes as no surprise, as it is in line with a common trade-off between purely Bayesian methods and optimization-centric ones, which enjoy higher scalability at the cost of harder uncertainty quantification. To conclude, we are grateful to the reviewer for raising these important conceptual points, which we will clarify further in the next version of the paper.
> * The reviewer asks about scalability. The method promises to be scalable to large datasets due to the criterion form seen in Equation (3), which lends itself to easy differentiation (under differentiability assumptions for $h$, which are independent of our method) and stochastic gradient optimization. Of course, as the data size increases, it is necessary to increase the parameter $T$ (truncation threshold) and/or $N$ (MC samples) to make the criterion representative of the sample (e.g., to ensure that each observation appears in the sample at least once), but linear scaling of either parameter suffices for that purpose.
> * The reviewer asks about the interpretability and restrictiveness of the criterion, e.g. in terms of data partitions. We highlight that this latter aspect is not applicable to our setting, as the data is modeled via a DP indirectly through a posterior expectation, but learning happens through standard optimization over parameters. Hence, the usual DP clustering is absent from this framework. In terms of restrictiveness, the nonparametric DP posterior choice ensures full support over the space of data-generating measures, but still preserves tractability due to its posterior and predictive characterizations. Notice that more general choices could be considered (e.g., the PY process), but tractability would be hindered and not much would be gained, for instance, in the presence of continuous data.
>
> Finally, we’d like to address the last limitation mentioned by the reviewer:
> * The reviewer indicates that our theoretical analysis focuses on the predictive risk but not on parameter estimation. Preliminarily, we would like to point out that Theorem 3.6 indeed deals with asymptotic convergence of the estimated parameter to its true counterpart. However, we thank the reviewer for the comment and agree with them that finite-sample analysis is still lacking for parameter estimation. This is due to the fact that, to obtain results in this direction, it is most likely necessary to specialize to particular cases of the loss function $h$ (e.g., quadratic loss for linear regression), which is the object of some of our ongoing research on the general method proposed in this paper.

---

> ### Comment · Area_Chair_vPn4 · 2024-08-11
> **discussion**
>
> Dear Reviewer p57R,
>
> We appreciate you submitting your review. The authors have provided replies to your comments. Could you kindly let us know if they have adequately addressed your points?
>
> Best regards, AC

---

> > ### Comment · Reviewer_p57R · 2024-08-12
> >
> > The authors have engaged with my concerns concerning the article, primarily about the unconventional use of Bayesian reasoning in a new context, and more pragmatic questions of scalability and the presentation of experiments. I appreciate the effort put in.
> >
> > I am happy to keep my score where it is.

---

### Official Review · Reviewer_Vivx · 2024-07-10

**Soundness:** 3
**Presentation:** 3
**Contribution:** 4
**Rating:** 7
**Confidence:** 4

**Summary:**

The authors use Dirichelet process and a smooth ambiguity aversion model to approximate the solution of risk minimization problems. They demonstate the consistency of their approach, along with finite sample guarantees. They additionally give a practical means of applying their procedure and apply the procedure to a number of simulated and real problems.

**Strengths:**

Background was very informative and enlightening. Despite being very technical in nature the paper was easy to follow. Based on the references given, the idea seems quite novel. Proofs related to results in section 3 were quite easy to follow (section 4 and appendix proofs not checked)

**Weaknesses:**

There are many small typos.
 - In Lemma 3.2 the supremum of \gamma^* is not needed as gamma^* does not depend on t. The statement in the proof is the supremum over gamma(t).
- For consistency with lemma 3.2, in theorem 3.3 the supremum over gamma(t) should probably be replaced by gamma^* (or gamma(t) should be added back into lemma 3.2).
 - In line 181 I don't think the (ii) should be there.

There are some minor problems.
 - The introduction of the dependency of phi on n makes a small appearence. This idea is very interesting but I don't think it is emphasised enough. Just a couple of sentences explaining why you'd want to do such a thing would be nice.
 - In line 221 a form of phi is assumed. I assume this is supposed to be phi_n and not phi. If it is supposed to be phi I'm unsure what form is being assumed, and if it is supposed to be phi_n it doesn't seem like it is assumed as section 4 seems to use phi.

And there are some reasonably major (but easily fixable) problems.
- On first read it is confusing how exactly the distance between the theoretic risk and the risk computed WRT to p_0 enters into lemma 3.2. Seeing the full statement I understand why the full bound isn't included in the main text but perhaps this discussion can be reworded.
- It is unclear if theorem 4.4 is using the same assumptions as lemma 4.3, or just the form of C_T that is given in lemma 4.3.
- I think theorem 3.3 needs to assume that M_phi < infinity as it is moved to the otherside of an equation in the proof. I think the proof needs more discussion about the case of gamma^*=infinity as on first read it looks like it moved to another side of the equation in the proof. gamma^*=infinity seems fine as thne the bound is triival. These cases are possible as taking K=1 and phi(t) = (t-1)^(1/3) has M_phi = gamma^*=infinity.  Seems like everything can be infinite in lemma 3.2 as then the bound is trivial.

**Questions:**

In theorem 3.4 it is assume that phi_n converges uniformly to the identity function. To me this seems like assuming that curvature of phi_n tends to 0 indictating that as n goes to infinity there is no ambiguity aversion. However, uniform convergence to the identity function is much a much stonger condition than curvature going to 0. For example convergence to 2 times the identity function still has curvature going to 0. Am I wrong in my reasoning behind convergence to the identity function? If not, is it possible to update assumption (3) in theorem 3.4 to a more general curvature to 0 assumption? It seems like the without much modification it can be instead assumed that phi_n converges to some scalar multiple of the idenity function.


How do the assumptions on theorem 3.3,3.4 guarantee that theta_n and theta_* exist? As is isn't assume that h is continuous or Theta is compact it seems like the argmin of V and R can be empty. From the proof of theorem 3.3 it seems like if theta_* doesn't exist it can be replaced by any arbitrary theta.

Another question. The uniform boundedness of h is strongly relied on to get the generate the results. Is it possible/are there methods available to remove this assumption? Classic problems like least squares regression over an unbounded domain will have h unbounded. This often isn't all that bad as assuming compactness of the domain is fine, but the bounds given here depend strongly on the size of the compact set used to restrict the problem to.

**Limitations:**

-

---

> ### Author Rebuttal · Authors · 2024-08-05
>
> We thank the reviewer for the time spent reviewing our work and for the insightful comments, which we will incorporate in the next version of our paper by fixing the typos/mistakes pointed out by the reviewer and by emphasizing the points brought up in the review.
>
> As for the minor problems pointed out by the reviewer, we’d like to address them as follows:
> * Better explanation of the dependence of $\phi$ on $n$: The paper includes a relatively careful explanation of this choice and its interpretation, as collected in Remark 3.7. We will however try to emphasize it more in the next version of the paper.
> * Specific form of $\phi$ in line 221: The reviewer spotted a typo ($\phi$ instead of the correct version $\phi_n$) which creates confusion. We thank them for pointing this out and we will fix it in the next version of the paper. As for our omission of the dependence on $n$ also in the subsequent sections, we have made this choice because such sections deal with approximation and gradient optimization of the criterion for a _fixed_ sample size $n$. In contrast, the dependence of $\phi$ on $n$ is only relevant as $n$ varies to ensure proper convergence results.
>
> As for the more serious yet fixable problems pointed out by the reviewer, we’d like to address them as follows:
> * The reviewer states that “On first read it is confusing how exactly the distance between the theoretic risk and the risk computed WRT to p_0 enters into lemma 3.2.” We agree with the comment and move footnote 8 to the main body, so to make this relation more evident.
> * The reviewer states that “It is unclear if theorem 4.4 is using the same assumptions as lemma 4.3, or just the form of C_T that is given in lemma 4.3.” We thank the reviewer for pointing this out, which gives us the opportunity to clarify the result. The exact form of $C_T$ can be found in the proof of Lemma A2 on page 23. In Theorem 4.4 we maintain the same assumptions as in Theorem 4.3, with the additional requirement that $C_T$ (defined as above) is bounded above as $T$ varies. We will make sure to clarify these points in the next version of the paper.
> * As for the boundedness of $M_\phi$ and $\gamma_\phi^*$, the reviewer is right in pointing out that our treatment silently assumes both quantities to be finite (as it is the case with the exponential functional form for $\phi$ proposed in the paper). Because in practice these assumptions are easily satisfied, we will explicitly mention them in the next version of the paper.
>
> As for the questions asked by the reviewer, we would like to address them as follows:
> * The reviewer asks whether assuming the uniform convergence of $\phi_n$ to the identity is necessary, or whether it can be relaxed with convergence to some multiple of the identity. We point out that, for the sake of optimization, any affine transformation of $\phi_n$ would yield the same optimized parameter. Hence, assuming that $\phi_n$ converges uniformly to the identity is without loss of generality (and, in one form or the other, needed for the proof of Theorem 3.4), because $\phi_n$ can always be normalized and centered to obtain the desired convergence. Notice that this is exactly the reason why we choose $\phi_n(t) = \beta_n \exp(t/\beta_n) - \beta_n$ as a special functional form, even though $\phi_n(t) = \exp(t/\beta_n)$ would be equivalent (optimization-wise) for all $n$.
> * The reviewer asks whether the existence of theoretical and empirical minimizers is ensured by our assumptions. We thank the reviewer for pointing out this lack of clarity, which we will resolve in the next version of the paper. The existence of minimizers is not a consequence of our assumptions on $h$ or $\Theta$, but rather an assumption itself – we assume the optimization problem to be well posed. In practice, many loss functions (e.g., convex) will easily ensure the existence of some optimizer (both theoretical and for our criterion, which is convex if the $h$ is convex).
> * The reviewer asks about the necessity of $h$ being uniformly bounded for our results. This is a good point, which we are currently tackling in follow-up research on the method we propose in this paper. Indeed, we believe that, specializing $h$ to a well-behaved loss like the quadratic loss used in linear regression, might yield finite sample and asymptotic guarantees even in the unbounded case (e.g., under appropriate tail conditions for the data-generating process $p_\star$). Being this paper our first, foundational contribution to this topic, we kept our analysis as general as possible in terms of loss function choice, which resulted in the need for stronger assumptions in our theoretical analysis. Nevertheless, as mentioned, the really good point made by the reviewer is currently being investigated, and we hope to have new results in that direction soon.

---

> ### Comment · Area_Chair_vPn4 · 2024-08-11
> **discussion**
>
> Dear Reviewer Vivx,
>
> Thank you for completing your review. The authors have responded to your feedback. Could you please let us know if their responses sufficiently address your concerns?
>
> Best regards, AC

---

> > ### Comment · Reviewer_Vivx · 2024-08-11
> >
> > Yes, the response sufficiently addresses my concerns.

---

> > > ### Author Response · Authors · 2024-08-11
> > >
> > > Dear Reviewer,
> > >
> > > We thank you again for the time spent reading and commenting our work, and remain at your disposal should any further question arise.
> > >
> > > Best regards,
> > >
> > > The Authors

---

### Official Review · Reviewer_7wnT · 2024-07-10

**Soundness:** 3
**Presentation:** 4
**Contribution:** 3
**Rating:** 7
**Confidence:** 3

**Summary:**

This paper introduces a new robust risk criterion that integrates concepts from Bayesian nonparametric theory, specifically the Dirichlet process, and a recent decision-theoretic model of smooth ambiguity-averse preferences. They show the relationships between their criterion and traditional regularization methods for empirical risk minimization, including Ridge and LASSO regressions. They also provide theoretical gurantee for the robust optimization and propose tractable approximations of the criterion.

**Strengths:**

This is a solid paper that makes a good and exciting theoretical contribution. The idea of incorporating Dirichlet prior into distributionally robust is novel, and well-suited for the problem. The empirical section is adequate.

**Weaknesses:**

*

**Questions:**

* I have a question about the coefficient in the prior. What will happen if we adopt the empirical Bayesian idea into your algorithm, i.e. estimate the coefficient in the prior through data? Will this cause degradation of the model？

**Limitations:**

The authors adequately addressed the limitations of their work.

---

> ### Author Rebuttal · Authors · 2024-08-05
>
> We thank the reviewer for the time spent reviewing our work and for the insightful comment, which we will incorporate in the next version of our paper by emphasizing its main point.
>
> Specifically, we believe that the reviewer makes a good point that, at first sight, it could be desirable to estimate the concentration parameter $\alpha$ from the data using empirical Bayes techniques. In principle, this solution could be viable by exploiting the exchangeable partition probability function (EPPF) of the Dirichlet Process, which can be read as the likelihood of observing the data clusters seen in the samples, given the parameter $\alpha$. Then, one could maximize the log-EPPF (available in closed form,  see Equation 2.19 in [1]) and obtain an empirical estimate of $\alpha$. However, unless the data contain ties (i.e., the data-generating distribution is assumed to have a discrete component, which is often unlikely in practical applications), this will lead to a degenerate $\alpha = +\infty$ solution – this is because the data is trivially partitioned into $n$ clusters, each with one component. One could further try to solve this issue by first applying some off-the-shelf clustering method to the data set, then maximizing the resulting “approximate EPPF”. While this option is interesting, it requires one further layer of computation, which adds to the overall cost of the optimization procedure. Instead, for practical purposes, a simple yet efficient approach might be to select the parameter value based on classical out-of-sample validation strategies. While this last solution is less grounded in the theory of Dirichlet Processes, we believe that it will be more beneficial in practice, as it is more aligned with the explicit aim of optimization-based machine learning methods that aim to maximize predictive performance.
>
>
> *[1] Pitman, J. (2006). Combinatorial stochastic processes: Ecole d'eté de probabilités de Saint-Flour XXXII-2002. Springer.*

---

> > ### Comment · Reviewer_7wnT · 2024-08-11
> >
> > Thank you for your rebuttal. I remain positive about the paper.

---

> > > ### Author Response · Authors · 2024-08-11
> > >
> > > Dear Reviewer,
> > >
> > > We thank you again for the time spent reading and commenting our work, and remain at your disposal should any further question arise.
> > >
> > > Best regards,
> > >
> > > The Authors

---

### Official Review · Reviewer_uUEe · 2024-07-12

**Soundness:** 3
**Presentation:** 3
**Contribution:** 3
**Rating:** 5
**Confidence:** 1

**Summary:**

This paper proposes a novel robust optimization criterion for training machine learning and statistical models by combining Bayesian nonparametric theory and smooth ambiguity-averse preferences, addressing distributional uncertainty to improve out-of-sample performance. The authors demonstrate theoretical guarantees for their method and show its practical implementation and effectiveness through tasks using simulated and real datasets.

**Strengths:**

The theoretical aspects of this paper are not my research area, so I am not equipped to assess the strengths and weaknesses of the theoretical contributions. However, the proposed method and its practical implications appear to be well-founded and promising.

**Weaknesses:**

The theoretical aspects of this paper are not my research area, so I am not equipped to assess the strengths and weaknesses of the theoretical contributions. However, the proposed method and its practical implications appear to be well-founded and promising.

**Questions:**

NO.

**Limitations:**

Discussed in Section 6.

---

> ### Author Rebuttal · Authors · 2024-08-05
>
> We thank the reviewer for the comments and time spent reviewing our work. If any further questions should come up during the next phases of the reviewing process, we will be happy to engage in further discussion with the reviewer.

---

### Official Review · Reviewer_DqoX · 2024-07-23

**Soundness:** 3
**Presentation:** 2
**Contribution:** 2
**Rating:** 5
**Confidence:** 2

**Summary:**

The paper proposes a Bayesian method for optimizing stochastic objectives under a given finite sample. More precisely, The paper assumes the Dirichlet Process for data generation and then it proposes to optimize the mean of the stochastic objective over posterior distributions given the sample. The paper proves asymptotic properties of the proposed objective. The experimental results in the appendix show that the proposed method works better than the standard L2 regularization or no regularization on linear regression tasks over simulation and real data sets.

**Strengths:**

The problem is quite relevant. The paper proves basic asymptotic desirable properties. The experimental results show the superiority of the proposed method to the standard L2 regularization-based methods.

**Weaknesses:**

One of my concerns is the proposed method's high computational cost. However, I don’t think it is critical. When the sample size is large enough, the standard ERM would work well. The proposed method, though it shows asymptotic convergence properties, would work better when the sample size is small. On the other hand, it would need more technical improvement when the sample space is high-dimensional.

**Questions:**

Is it difficult to obtain finite-sample error bound like PAC or statistical learning theory? (or any related work?)

---

> ### Author Rebuttal · Authors · 2024-08-05
>
> We thank the reviewer for the time spent reviewing our work and for the insightful comments, which we will incorporate in the next version of our paper by emphasizing the points brought up by the reviewer.
>
> Specifically, we would first like to address the weakness, kindly pointed out by the reviewer, related to the computational cost of the method. We agree with the observation that the method is best suited for small-to-moderate sample sizes, as is true for any Distributionally Robust Optimization (DRO) method. In fact, distributional uncertainty is likely to affect optimization only when a relatively small sample does not allow to accurately approximate the data generating process. However, as the sample size increases, distributional uncertainty becomes less of an issue and standard ERM methods suffice for accurate parameter estimation and stable predictive performance. Nevertheless, while this is an important observation for the practical usefulness of any DRO method, we believe that sample size is not per se a computational problem for our method. Indeed, looking at Equation (3) in the paper, it is clear how in practice the criterion is estimated by cheaply sampling from the DP predictive and appropriate weight distributions. Therefore, while a larger sample size might require a higher value of $T$ (truncation threshold) and/or $N$ (number of MC samples) to make the criterion more representative of the data set (e.g., ensure that each data point is sampled at least once with high probability), gradient computations are easily vectorized and lead to very efficient mini-batch SGD updates akin to plain ERM methods. The same holds true for the dimensionality of the data and/or parameter space, which affects computational cost in the same way as it does for standard ERM.
>
> Secondly, we’d like to address the reviewer’s question about the possibility of obtaining finite-sample performance guarantees. We point out that this is precisely the goal of Theorem 3.3, which relates finite sample bounds on the predictive performance parameter estimator $\theta_n$ to the classical $\sup$ distance between the ERM and true risk. The latter, in turn, allows us to obtain finite-sample probabilistic bounds via conditions on the complexity (VC dimension, metric entropy, etc.) of the loss function class, as done in standard learning theory. As we point out in the paragraph after Theorem 3.3, our paper does not discuss any specific result of this kind as they are (1) standard in the literature and (2) can be immediately plugged in from classical textbook formulations.

---

> ### Comment · Area_Chair_vPn4 · 2024-08-11
> **discussion**
>
> Dear Reviewer DqoX,
>
> Thank you very much for submitting your review report. The author(s) have posted responses to your review. Could you kindly provide comments on whether your concerns have been adequately addressed?
>
> Best regards, AC

---

### Comment · Area_Chair_vPn4 · 2024-08-11
**some comments**

Dear Author(s),

I would like to invite you to clarify some points which I am uncertain about. Thank you very much!

1. It appears that Propositions 2.1 and 3.1 are well-established, though finding a convenient source for citation might be challenging. I wonder if it would be appropriate to mention this in the revised paper.

2. In comparison with the definition in (1), I wonder if it might be more straightforward to directly link V-hat (as defined in (3)) to ϕ(Rp∗(θ))—the quantity on the right-hand side of the equation in Proposition 3.1—for the analysis. Is there any particular challenge or specific reason for using (1)?

---

> ### Author Response · Authors · 2024-08-11
>
> Dear Area Chair,
>
> Thank you for your insightful comments and careful reading of our submission. We’d like to address your questions as follows:
>
> 1. We agree with your observation regarding the need for a discussion on the original sources for results similar to Propositions 2.1 and 3.1, and we will incorporate this in the next version of the paper. Specifically, for Proposition 3.1, we will move the citation of [1], currently in the appendix, to the main body as a standard reference on DP weak convergence to the true generating distribution. Regarding Proposition 2.1, we acknowledge that pinpointing the exact origin of the classical interpretation of Ridge regression as Bayesian linear regression with a *parametric* standard normal prior on the coefficients is challenging. However, as noted after Proposition 2.1, our result offers a novel Bayesian interpretation: by taking an optimization-centric view and placing a *nonparametric* prior directly on the data-generating distributions (instead of on the coefficients, which we optimize), we also arrive at Ridge regularization. We believe this result highlights the fundamental connections between optimization, decision theory, and Bayesian inference, and we will ensure this is clearly explained in the revised paper.
>
> 2. We also agree that our paper currently lacks an explicit discussion of our reasoning for relating $\hat V$ to $V_{\boldsymbol\xi^n}$ instead of directly to $\phi(\mathcal R_{p_\star}(\cdot))$. We will address this in the next version of the paper with the following clarifications. As you noted, one's ultimate goal may be to ensure the convergence of $\hat V$ to $\phi(\mathcal R_{p_\star}(\cdot))$, which involves three layers of approximation: from the finite sample size $n$, from the random measure truncation threshold $T$, and from the number of MC samples $N$. In Section 3, we address the first layer by studying the convergence of $V_{\boldsymbol\xi^n}$ to $\phi(\mathcal R_{p_\star}(\cdot))$. In Section 4, we focus on the latter two layers, *given a fixed sample size approximation determined* by $n$, by studying the convergence of $\hat V$ to $V_{\boldsymbol\xi^n}$. Thus, within this logical chain, $V_{\boldsymbol\xi^n}$ serves as a bridging quantity between $\hat V$ and $\phi(\mathcal R_{p_\star}(\cdot))$. The results from Sections 3 and 4 can then be combined to directly establish the convergence of $\hat V$ to $\phi(\mathcal R_{p_\star}(\cdot))$, which involves choosing $T$ and $N$ as functions of $n$ to ensure that the right-hand side of the first equation in the statement of Lemma 4.3 converges to zero. We will emphasize this crucial point in our next revision.
>
> Thank you again for your valuable feedback,
>
> The Authors
>
>
> *[1] S. Ghosal and A. Van der Vaart. Fundamentals of nonparametric Bayesian inference, volume 44. 378 Cambridge University Press, 2017.*

---

> > ### Comment · Area_Chair_vPn4 · 2024-08-12
> > **thank you very much**
> >
> > I appreciate your kind response.

---

### Decision · Program_Chairs · 2024-09-25

**Decision:**

Accept (poster)

**Comment:**

The paper addresses the challenge of optimizing machine learning models under distributional uncertainty. The authors propose a novel method that integrates Bayesian nonparametric theory, specifically the Dirichlet Process (DP), with distributionally robust optimization (DRO). Finite-sample probabilistic guarantee on the excess risk is derived. The asymptotic vanishing of the excess risk and the convergence of the finite-sample optimal value are established. The paper's theoretical contributions are considered solid, with the proofs being clear and technically sound (see the review and comments by Reviewers 7wnT and Vivx). However, the experimental results, while promising, are briefly discussed in the main text (see the review and comments by Reviewer p57R). Some other concerns include: computational expense, especially for high-dimensional data or large sample sizes (see the review and comments by Reviewer DqoX); could better explain the implications of placing a prior on the data-generating process rather than on model parameters (see the review and comments by Reviewer p57R); minor typos (see the review and comments by Reviewer Vivx).